# NHR-14 loss of function couples intestinal iron uptake with innate immunity in *C. elegans* through PQM-1 signaling

Malini Rajan[1,2†], Cole P Anderson[2,3†], Paul M Rindler[1,2†], Steven Joshua Romney[1,2†], Maria C Ferreira dos Santos[1,2], Jason Gertz[3,4], Elizabeth A Leibold[1,2,3]*

[1]Department of Medicine, Division of Hematology, University of Utah, Salt Lake City, United States; [2]Molecular Medicine Program, University of Utah, Salt Lake City, United States; [3]Department of Oncological Sciences, University of Utah, Salt Lake City, United States; [4]Huntsman Cancer Institute, University of Utah, Salt Lake City, United States

**Abstract** Iron is essential for survival of most organisms. All organisms have thus developed mechanisms to sense, acquire and sequester iron. In *C. elegan*s, iron uptake and sequestration are regulated by HIF-1. We previously showed that *hif-1* mutants are developmentally delayed when grown under iron limitation. Here we identify *nhr-14*, encoding a nuclear receptor, in a screen conducted for mutations that rescue the developmental delay of *hif-1* mutants under iron limitation. *nhr-14* loss upregulates the intestinal metal transporter SMF-3 to increase iron uptake in *hif-1* mutants. *nhr-14* mutants display increased expression of innate immune genes and DAF-16/FoxO-Class II genes, and enhanced resistance to *Pseudomonas aeruginosa*. These responses are dependent on the transcription factor PQM-1, which localizes to intestinal cell nuclei in *nhr-14* mutants. Our data reveal how *C. elegans* utilizes nuclear receptors to regulate innate immunity and iron availability, and show iron sequestration as a component of the innate immune response.
DOI: https://doi.org/10.7554/eLife.44674.001

*For correspondence:
betty.leibold@genetics.utah.edu

†These authors contributed equally to this work

**Competing interests:** The authors declare that no competing interests exist.

## Introduction

Iron is an essential nutrient required by nearly all organisms for biological processes including DNA replication, mitochondrial respiration, and oxygen transport (*Anderson et al., 2012*). As a redox active element, iron also has a role in the production of reactive oxygen species (ROS), which can be deleterious to cellular proteins, DNA and membranes. In humans, tight regulation of cellular iron content is essential, as iron deficiency impairs neurological development in infants and causes anemia in adults, while iron overload leads to cardiomyopathy, diabetes mellitus, and neurodegenerative diseases (*Fleming and Ponka, 2012*). Excess iron also predisposes humans to infectious diseases, and limiting iron availability to microbes is a critical host innate immune response (*Lopez and Skaar, 2018*; *Weiss and Carver, 2018*). Given the dual nature of iron in healthy and diseased states, it is critical for organisms to maintain cellular iron concentration in a narrow physiological range.

*C. elegans* has emerged as a useful model for studying iron metabolism. In addition to its genetic tractability, ease of maintenance and susceptibility to human pathogens, key proteins involved in mammalian iron uptake, storage and efflux are conserved in *C. elegans* (*Anderson and Leibold, 2014*). *C. elegans* express the divalent metal transporter SMF-3 (*Au et al., 2009*; *Bandyopadhyay et al., 2009*; *Romney et al., 2011*), the iron-storage protein ferritin (FTN-1, FTN-2) (*Gourley et al., 2003*; *Kim et al., 2004*; *Romney et al., 2008*; *Valentini et al., 2012*) and the iron

export protein ferroportin (FPN-1.1–3), which are orthologous to human DMT1 (SLC11A2, NRAMP2), ferritin-H and -L subunits (FTH1, FTL) and ferroportin (SLC40A1), respectively. Iron is transported across the apical enterocyte membrane by SMF-3/DMT1 and is used within the enterocyte for the metalation of iron-containing proteins and by mitochondria for Fe-S cluster biosynthesis (*Braymer and Lill, 2017*; *Rouault, 2015*). Iron not utilized is stored in ferritin within the enterocyte or exported across the basolateral enterocyte membrane by ferroportin/FPN-1.1. Ferroportin is the sole iron exporter in mammals (*Ward and Kaplan, 2012*), unlike *C. elegans* that express three ferroportin orthologs (FPN-1.1–3) that are not well characterized. Like mammals, intestinal iron absorption in *C. elegans* is upregulated during iron deficiency by HIF-transcriptional activation of *smf-3/DMT1* (*C. elegans*, HIF-1; human, HIF-2α, EPAS1) (*Mastrogiannaki et al., 2009*; *Romney et al., 2008*; *Shah et al., 2009*). One difference between iron sequestration in *C. elegans* and vertebrates is that during iron limitation *C. elegans* ferritin is transcriptionally repressed by HIF-1 (*Ackerman and Gems, 2012*; *Romney et al., 2011*), while vertebrate ferritin is translationally repressed by the iron-regulatory proteins (IRPs) (*Anderson et al., 2012*). In addition to iron, *ftn-1* is regulated by the insulin/IGF-like (IIS) pathway (*Ackerman and Gems, 2012*), the REF-1 transcription factor HLH-29 (*Quach et al., 2013*), and by the serum/glucocorticoid-regulated kinase (SGK-1) to regulate lipid and iron homeostasis (*Wang et al., 2016*). The regulation of iron uptake by SMF-3 and iron sequestration by ferritin ensures that cells acquire adequate iron to satisfy their needs while limiting iron toxicity. Precise maintenance of cellular iron homeostasis is underscored by the shortened life span of *C. elegans* exposed to high iron (*Gourley et al., 2003*; *Valentini et al., 2012*).

Iron is required in the establishment of most pathogen infections. Pathogens use a variety of mechanisms to acquire iron from the host, one of which is the synthesis and release of siderophores that bind iron, allowing its uptake by bacteria (*Palmer and Skaar, 2016*). In response to pathogens, hosts have evolved strategies to limit iron availability that include the production of metal sequestering proteins, the sequestration of iron in ferritin and the regulation of metal transporter function (*Lopez and Skaar, 2018*; *Wessling-Resnick, 2015*). *C. elegans* has provided a valuable genetic model to study host-pathogen interactions because many human pathogens, such as *Pseudomonas aeruginosa*, *Salmonella enterica* and *Staphylococcus aureus*, cause lethal intestinal infections in *C. elegans* (*Cohen and Troemel, 2015*). Although *C. elegans* lack an adaptive immune system, these organisms rely on evolutionarily conserved innate immune systems, such as the p38-mitogen-activated protein kinase (MAPK), insulin and transforming growth factor-β (TGF-β) signaling pathways, for sensing and resolving bacterial infections (*Irazoqui et al., 2010*; *Kim and Ewbank, 2015*). Similar to vertebrates, *C. elegans* use diverse mechanisms to withhold iron and other metals during pathogen infections. FTN-2 provide resistance to *S. aureus* and *E. coli* LF82 infections (*Simonsen et al., 2011*) and SMF-3 provides resistance to *S. aureus* infections (*Bandyopadhyay et al., 2009*). In addition, *C. elegans* respond to pyoverdine, an iron siderophore produced by *P. aeruginosa*, by upregulation of a mitochondrial surveillance pathway that is triggered by pyoverdine-induced mitochondrial damage (*Kang et al., 2018*; *Kirienko et al., 2015*; *Tjahjono and Kirienko, 2017*). These studies highlight iron limitation as an important component of innate immune response in controlling pathogen infections.

In this report, we identify NHR-14, a nuclear receptor (NR) as a transcriptional repressor of SMF-3-dependent iron uptake and the innate immune response. Loss of *nhr-14* promotes the nuclear localization of PQM-1, which binds to GATA-like DAF-16-associated elements (DAEs) in the *smf-3* promoter to activate *smf-3* transcription. *nhr-14* mutants exhibit enhanced resistance to *P. aeruginosa* infection that in part requires *smf-3* as well as other innate immune response genes. Our data implicate NHR-14 as a repressor of iron uptake and the transcriptional innate immune response mediated by PQM-1, showing iron sequestration as an important component of innate immunity in *C. elegans*.

## Results

### Loss of NHR-14 rescues low iron growth of *hif-1(ia4)* mutants

We previously reported that *hif-1(ia4)* mutant worms are developmentally delayed when grown under iron limiting conditions using the iron chelator, 2,2-bipyridyl (BP), and that this delay was suppressed by elevating intracellular iron by reducing *ftn-1* and *ftn-2* expression (*Romney et al., 2011*).

To discover novel genes involved in iron homeostasis, a suppressor screen was performed to identify mutations that rescued the low iron phenotype observed in *hif-1(ia4)* null animals. The *ia4* mutation is a 1,231 bp deletion of the second, third, and fourth exons that deletes much of the bHLH and PAS domains in HIF-1, and likely results in complete loss of *hif-1* function (*Jiang et al., 2001*). This screen led to the isolation of several suppressor mutants, of which *nhr-14(qa6909)* and *nhr-14 (qa6910)* were identified as strong suppressors of the *hif-1(ia4)* low iron phenotype (*Figure 1A*). The causative mutation within these lines was determined using whole genome sequencing and Hawaiian strain SNP mapping (see Materials and methods). Both the *qa6909* and *qa6910* mutations displayed a similar ~1 Mb region of the X chromosome that have a Hawaiian/Bristol reads ratio of close to zero (*Figure 1B*). Sequence analysis revealed that the *qa6909* and *qa6910* mutant strains possess distinct mutations within the nuclear-hormone receptor gene *nhr-14* (T01B10.4) (*Figure 1C*). DNA from *qa6909* mutants showed a G > A substitution that changes Trp49 to a STOP codon and DNA from *qa6910* mutants showed a G > A substitution that changes Gly33 to Arg (*Figure 1C*).

*nhr-14* is predicted to encode a protein homologous to the HNF4 family of nuclear receptors, which are transcription factors that regulate gene expression in response to nutritional, metabolic and environmental signals (*Taubert et al., 2011*). Amino acid sequence alignment between NHR-14 and human HNF4α showed a highly conserved N-terminal zinc finger DNA-binding domain (DBD) and a conserved C-terminal ligand-binding domain (LBD) (*Figure 1C*). The *qa6909* mutation (Trp49STOP) located within the DBD is predicted to result in a truncated protein that lacks a significant portion of the DBD and the entire LBD. The *qa6910* missense mutation (Gly33Arg), which is also located within the DBD, changes non-polar glycine to positively-charged arginine (*Figure 1C*). Gly33 is conserved in HNF4 receptors found in vertebrates and *Drosophila*, and it is likely that this substitution disrupts DNA binding.

To verify that mutations in *nhr-14* are responsible for rescuing the low iron developmental delay of *hif-1(ia4)* mutants, we crossed *nhr-14(tm1473)* mutants into the *hif-1(ia4)* background, and grew *hif-1(ia4); nhr-14(tm1473)* double mutants on low iron (NGM-BP) plates. The *tm1473* mutation is a 409 bp deletion of the third exon and intron that results in a premature stop codon and deletion of the ligand-binding domain that likely results in *nhr-14* loss of function. *hif-1(ia4); nhr-14(tm1473)* double mutants developed normally under iron limitation (*Figure 1D*), showing that loss of *nhr-14* function is responsible for the rescue observed in *qa6909* and *qa6910* mutant animals.

## NHR-14 is highly expressed in intestine and head neurons and is not regulated by iron

Given that the mutations in *nhr-14* were identified in a screen for mutants that rescued the low iron developmental delay of *hif-1(ia4)* mutants, we questioned whether *nhr-14* mRNA or protein is regulated by changes in iron status. We generated a transgenic line carrying a *nhr-14* fosmid transgene that utilizes the endogenous *cis*-elements and promoter of *nhr-14* to drive the expression of NHR-14 tagged with GFP and 3xFLAG. We validated NHR-14::GFP::FLAG as a functional replacement for endogenous NHR-14 by injecting *hif-1(ia4); nhr-14(tm1473)* double mutants with the *nhr-14* fosmid transgene. These transgenic worms displayed the *hif-1(ia4)* developmental delay phenotype when grown under iron limitation (*Figure 2—figure supplement 1*). Fluorescent imaging of NHR-14:: GFP::FLAG worms showed that it is highly expressed in intestinal nuclei and in cells in the head (*Figure 2A–C*). Western blot analysis showed a protein of ~80 kDa in worm lysates consistent with the predicted mass of NHR-14::GFP::FLAG (*Figure 2D*). No significant change in levels of NHR-14:: GFP::FLAG and *nhr-14* mRNA were observed in wild-type N2 worms grown in low iron (NGM-BP) or high iron (NGM- ferric ammonium citrate, FAC) compared to NGM (*Figure 2D–E*). Furthermore, nuclear localization of NHR-14::GFP::FLAG was not affected by iron or by BP (*Figure 2—figure supplement 2*). The enrichment of *nhr-14* in intestinal and head cells is consistent with intestinal and neuronal expression of *nhr-14* determined by single-cell RNA sequence and SAGE analyses and (*Cao et al., 2017*; *McGhee, 2007*).

## Loss of NHR-14 regulates iron uptake by SMF-3

To determine the mechanism through which *nhr-14(tm1473)* mutants rescued the low iron developmental delay of *hif-1(ia4)* mutants, we compared the transcriptional profiles of L4-stage *nhr-14 (tm1473)* mutants with wild-type N2 worms using RNA-seq. We identified 834 differentially

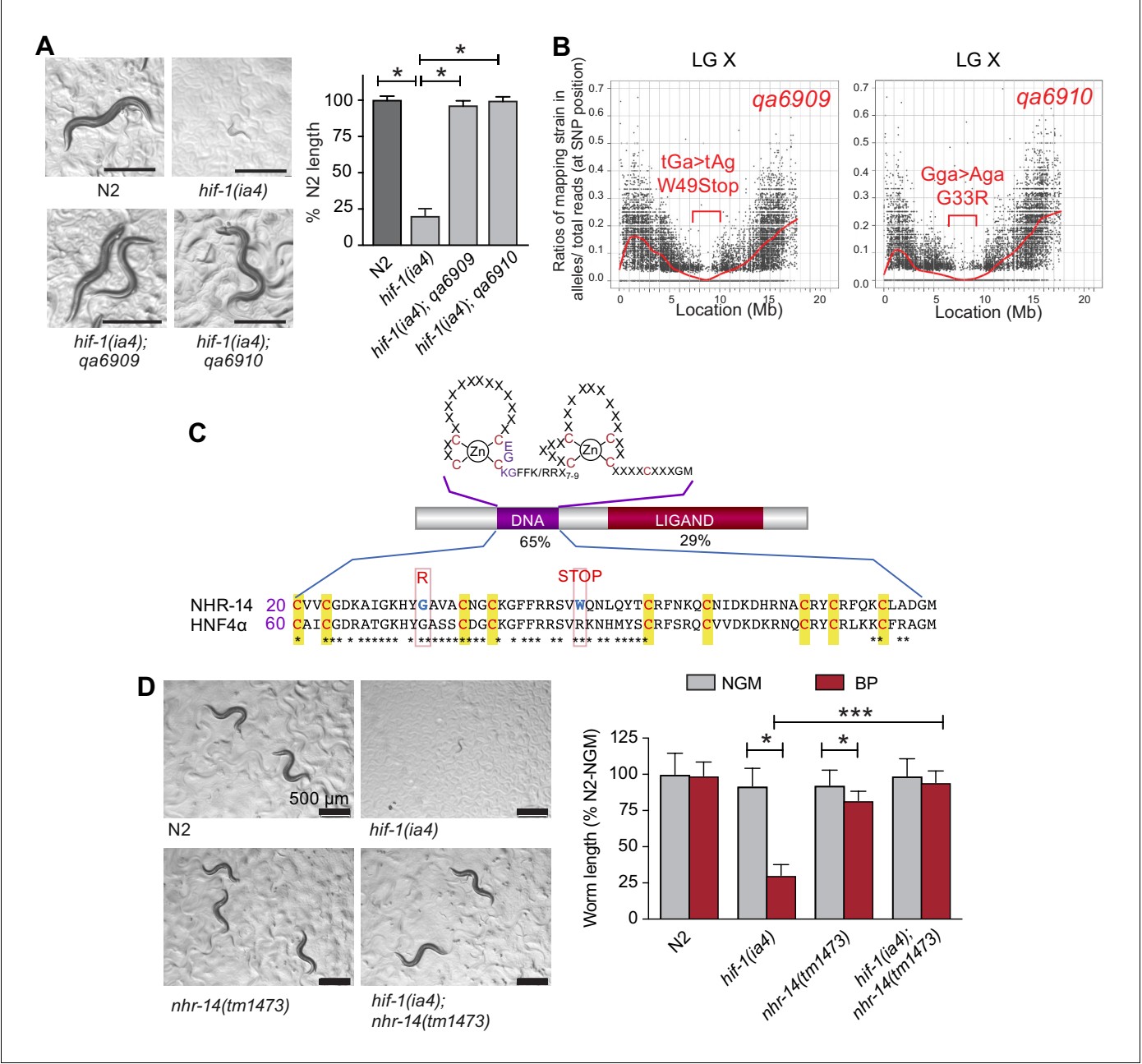

**Figure 1.** Mutations in *nhr-14* rescue the low iron developmental delay in *hif-1(ia4)* mutants. (**A**) Representative images of wild-type N2 worms, *hif-1(ia4)* mutants and *hif-1(qa6909)* and *hif-1(qa6910)* mutant alleles grown for 5 days on NGM plates containing 25 uM BP (bipyridyl, iron chelator). Worm length is expressed as a percentage of wild-type N2 length (n = 2 biological replicates with 10 worms per strain). Values are expressed as mean ± SEM and were compared by unpaired two-tailed Student's *t*-test, *p<0.05. (**B**) CloudMap SNP Mapping are shown as XY-scatter plots where the ratio of Hawaiian SNPs to Bristol SNPs is represented for the *qa6909* and *qa6910* mutations. (**C**) Protein alignment of the DNA-binding domains for human HNF4α and NHR-14 with locations of the *qa6909* and *qa6910* mutations (adapted from Figure 1 from *Antebi, 2006*, published under the terms of the Creative Commons Attribution License – CC-By 2.5 license; https://creativecommons.org/licenses/by/2.5/). Amino acid mutations are indicated in red. (**D**) Representative images of wild-type N2, *hif-1(ia4)*, *nhr-14(tm1473)* and *hif-1(ia4)*; *nhr-14(tm1473)* worms grown in NGM or NGM-25 uM BP (low iron) after 5 days. Worm length is expressed as a percentage of wild-type N2-NGM length (one experiment, n = 20 - 30 worms per strain). Values are expressed as mean ± SEM and compared by two-way ANOVA with Tukey's multiple comparisons test, *p < 0.05; ***p < 0.001. Scale bars, 500 um.
DOI: https://doi.org/10.7554/eLife.44674.002

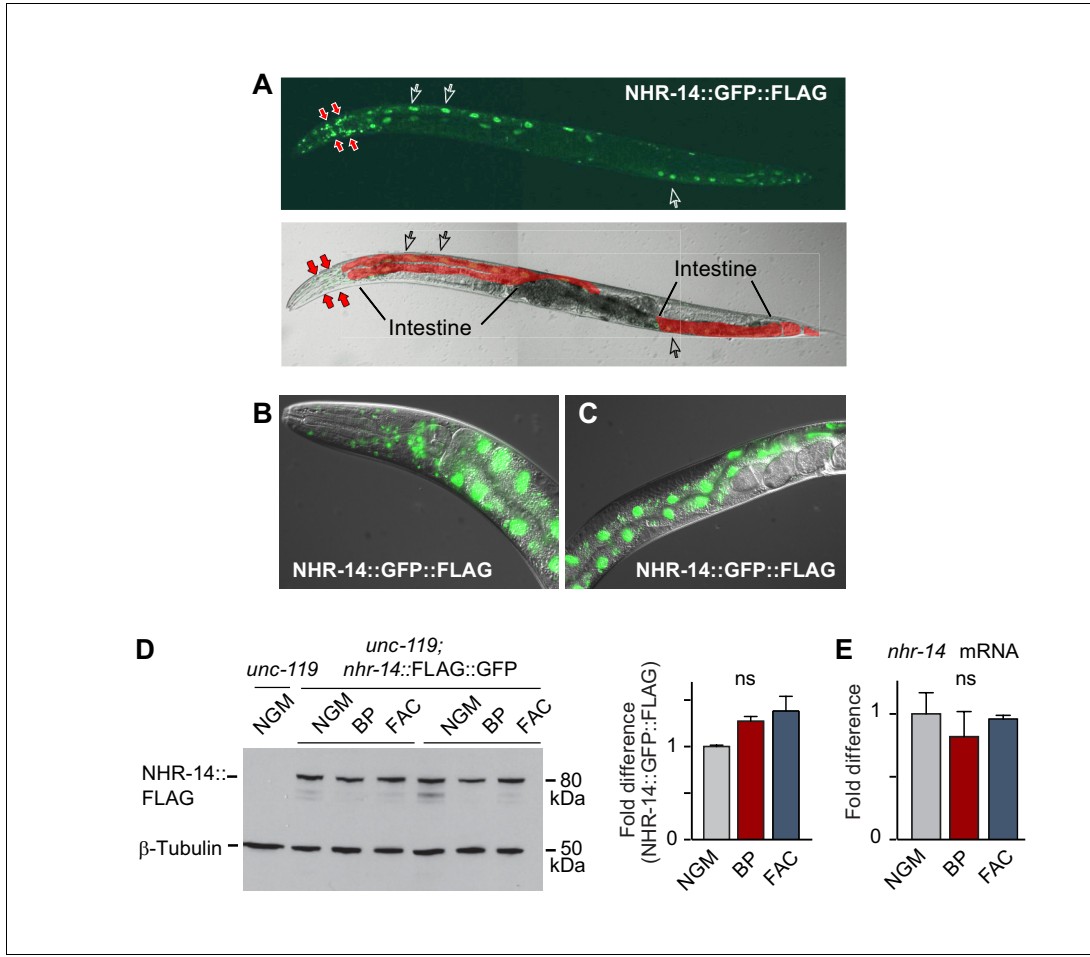

**Figure 2.** NHR-14 is highly expressed within intestinal and head cell nuclei and is not regulated by iron. (**A**) NHR-14::GFP::FLAG expression in young adult transgenic worms assessed by confocal microscopy (40x). Open arrows, intestinal nuclei; red arrows, head cells. (**B**) GFP expression of NHR-14::GFP::FLAG in cells in the (**B**) head and (**C**) intestine (60x). (**D**) Western blot analysis of NHR-14::GFP::FLAG non-injected worms (*unc-119*) versus NHR-14::GFP::FLAG worms grown in NGM, NGM-BP (low iron) and NGM-FAC (high iron) (two biological replicates are shown); right panel, quantification of NHR-14::GFP::FLAG expression (n = 3 biological replicates). The blot was simultaneously probed with FLAG and β-tubulin antibodies. (**E**) qPCR of *nhr-14* expression in wild-type N2 adults grown in NGM, NGM-BP and NGM-FAC normalized to N2 worms grown in NGM (n = 3 biological replicates). Values are expressed as mean ± SEM and are compared by unpaired two-tailed Student's *t* test; ns, not significant.

DOI: https://doi.org/10.7554/eLife.44674.003

The following figure supplements are available for figure 2:

**Figure supplement 1.** NHR-14::GFP::FLAG functions similarly to endogenous NHR-14.

DOI: https://doi.org/10.7554/eLife.44674.004

**Figure supplement 2.** Iron does not affect the nuclear localization of NHR-14::GFP::FLAG.

DOI: https://doi.org/10.7554/eLife.44674.005

expressed genes that changed at least two-fold compared to wild-type N2 worms at a 5% false discovery rate (FDR), 573 of which were upregulated and 261 downregulated in *nhr-14(tm1473)* mutants (*Figure 3—source data 1*). Notably, the primary intestinal iron transporter *smf-3* is one of the most highly upregulated genes in *nhr-14(tm1473)* mutants. This observation suggested increased iron uptake as a possible mechanism to explain rescue of *hif-1(ia4)* mutants under iron limitation.

We confirmed *smf-3* upregulation in *nhr-14(tm1473)* single mutants and in *hif-1(ia4); nhr-14 (tm1473)* double mutants compared to wild-type N2 worms, and *smf-3* downregulation in *hif-1(ia4)* mutants by qPCR (*Figure 3A*). Consistent with our previous study, *hif-1(ia4)* and *smf-3(ok1035)*

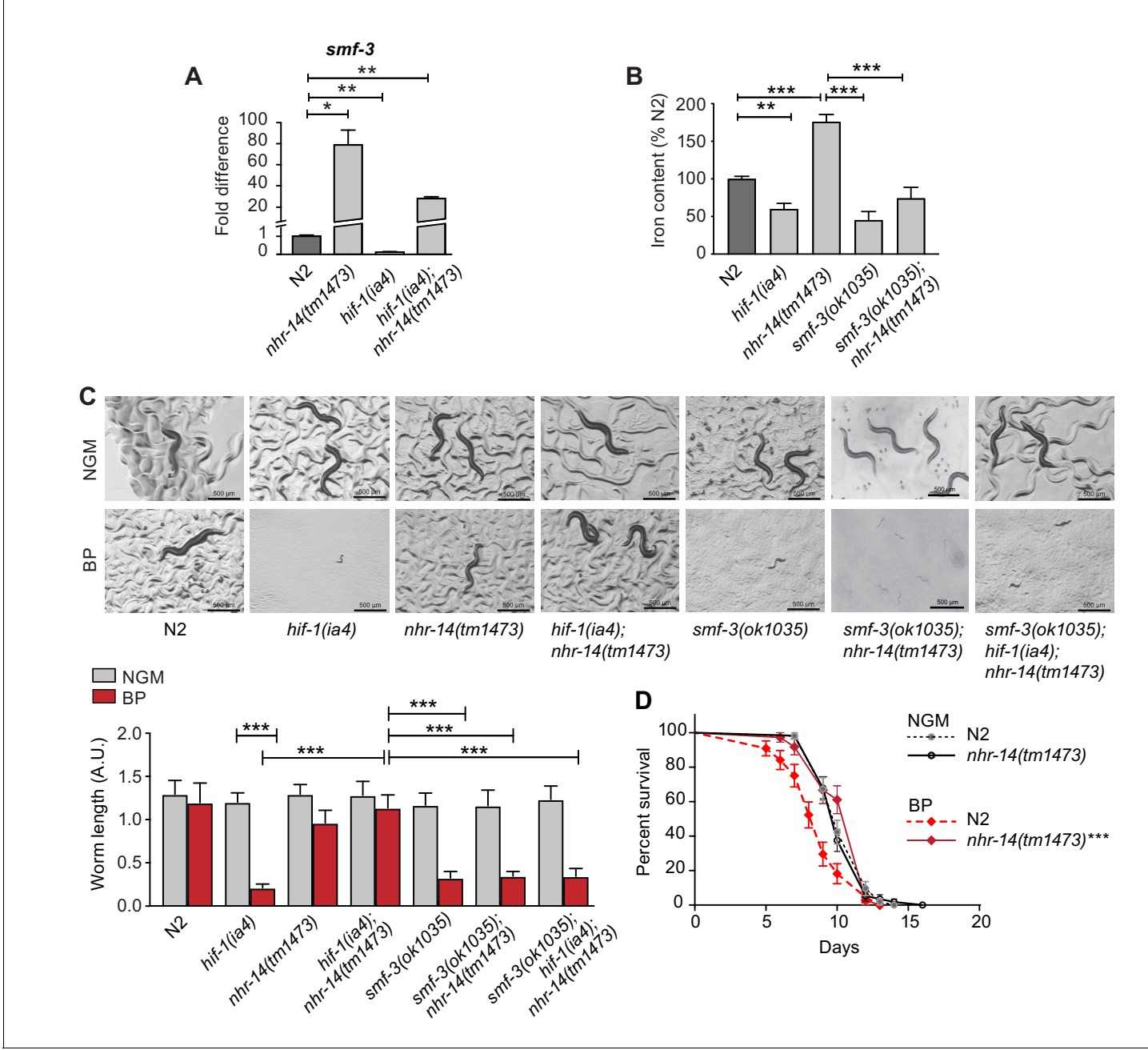

**Figure 3.** Loss of NHR-14 regulates iron uptake by SMF-3. (**A**) qPCR of *smf-3* expression measured in wild-type N2, *nhr-14(tm1473)* and *hif-1(ia4)* single mutants, and *hif-1(ia4); nhr-14(tm1473)* double mutants grown in NGM (n = 3 biological replicates). Values are expressed as fold difference compared to wild-type N2 worms. (**B**) Total iron content of wild-type N2 worms, *hif-1(ia4)*, *nhr-14(tm1473)* and *smf-3(ok1035)* single mutants, and *smf-3(ok1035); nhr-14(tm1473)* double mutants grown in NGM and quantified by ICP-OES (n = 3 biological replicates). For A-B, values are expressed as mean ± SEM, and compared by unpaired two-tailed Student's *t*-test for (**A**) and a two-way ANOVA with Tukey's multiple comparisons test for (**B**), *p<0.05, **p<0.01, ***p<0.001. (**C**) Representative images and quantitation of wild-type N2 worms and indicated mutant strains grown on NGM or NGM-BP (low iron) plates for 5 days. Data are combined from two biological replicates (n = 14–31 worms/strain). Values are expressed as mean worm length ± SEM and compared by two-way ANOVA with Tukey's multiple comparisons test, ***p<0.001. Scale bar, 500 um. (**D**) Survival analysis of wild-type N2 and *nhr-14 (tm1473)* worms grown in NGM and NGM-BP (low iron). Median survival time (MS): wild-type N2-NGM (MS = 10 days, n = 52); *nhr-14(tm1473)*-NGM (MS = 10 days, n = 56); wild-type N2-BP (MS = 9 days, n = 44); *nhr-14(tm1473)*-BP (MS = 12 days, n = 36), Log-rank Mantel-Cox test, ***p<0.001, wild-type N2-BP versus *nhr-14(tm1473)*-BP.

DOI: https://doi.org/10.7554/eLife.44674.006

The following source data is available for figure 3:

**Source data 1.** RNA-seq gene list of upregulated and downregulated *nhr-14(tm1473)* genes.
DOI: https://doi.org/10.7554/eLife.44674.007

mutants displayed reduced iron content compared to wild-type N2 worms (*Romney et al., 2011*) (*Figure 3B*). Iron content increased in *nhr-14(tm1473)* mutants consistent with increased *smf-3* expression, and decreased in *smf-3(ok1035); nhr-14(tm1473)* double mutants (*Figure 3B*). *smf-3 (ok1035)* single mutants, *smf-3(ok1035); nhr-14(tm1473)* double mutants and *smf-3(ok1035); hif-1 (ia4); nhr-14(tm1473)* triple mutants also displayed a developmental delay under iron limitation showing that *smf-3* upregulation in *nhr-14(tm1473)* mutants is required for the rescue of *hif-1(ia4)* mutants (*Figure 3C*). Given that the *smf-3(ok1035)* single mutant alone displays a developmental delay under iron limitation, this suggests that other genes in *nhr-14(tm1473)* mutants might also contribute to the rescue of *hif-1(ia4)* mutants. Further evidence for SMF-3-mediated iron uptake as a mechanism to explain the rescue of *hif-1(ia4)* mutants under iron limitation is the survival advantage of *nhr-14 (tm1473)* mutants grown in low iron (BP) as compared to wild-type N2 worms (*Figure 3D*). Collectively, these results are consistent with NHR-14 as a negative regulator of SMF-3-mediated iron uptake.

## PQM-1 acts downstream of NHR-14 to regulate *smf-3*

We next questioned whether NHR-14 regulates *smf-3* expression using transgenic worms expressing a GFP-H2B transcriptional reporter containing 1.5 kb of *smf-3* upstream regulatory sequences (P*smf-3::GFP-H2B*) (*Figure 4A*). This region harbors the 118 bp iron-dependent element (IDE) containing two hypoxia-response elements (HREs) and three GATA motifs (WGATAR) that are binding sites for HIF-1 and GATA proteins, respectively. We previously reported that P*smf-3::GFP-H2B* upregulation in intestine during iron deficiency is dependent on the *smf-3* IDE (*Romney et al., 2011*). To determine whether NHR-14 regulates *smf-3*, worms expressing P*smf-3::GFP-H2B* were fed empty vector control RNAi (Con) or *nhr-14* RNAi, and grown under normal (NGM), low iron (BP) or high iron (FAC) conditions, and GFP expression was assessed in L4 stage worms using the COPAS Biosort (*Figure 4B* and *Figure 4—figure supplement 1*). We verified rescue of the low iron developmental delay of *hif-1(ia4)* mutants by *nhr-14* RNAi (*Figure 4—figure supplement 2*). Consistent with our previous study (*Romney et al., 2011*), GFP expression increased with BP and decreased with FAC in control-RNAi fed worms, reflecting HIF-1 regulation (*Figure 4B*). GFP expression is further increased in *nhr-14*-RNAi fed worms with NGM, NGM-BP and NGM-FAC treatments, indicating that sequences within the *smf-3* promoter are important for NHR-14 regulation.

To identify transcription factor(s) binding to *smf-3* promoter sequences, we examined Wormbase data set containing genome-wide in vivo binding profiles for *C. elegans* transcription factors (*Niu et al., 2011*) (*Figure 4—figure supplement 3*). We found that the transcription factor PQM-1 showed significant association with the *smf-3* IDE (IV: 2618542..2619063; qValue $1.7e^{-32}$), suggesting that PQM-1 might be involved in *smf-3* activation in *nhr-14(tm1473)* mutants. PQM-1 associates with a GATA-like motif (TGATAAG) termed the DAF-16-associated element (DAE) found in the promoters of DAF-16/FoxO-Class II genes (*Tepper et al., 2013*). Class II genes are characterized as being downregulated by DAF-16 signaling (*Murphy et al., 2003*). Given ChIP-seq data showing that PQM-1 associates with the *smf-3* promoter, we tested the hypothesis that PQM-1 regulates *smf-3*. Single mutations in DAE 1, 2 or 3 motifs (T<u>G</u>ATAAG to T<u>C</u>ATAAG) were associated with reduced GFP expression in untreated and BP-treated worms compared to P*smf-3::GFP-H2B* worms carrying wildtype DAEs (*Figure 4C—figure supplement 1*). Endogenous *smf-3* expression was also reduced in *pqm-1(ok485)* mutants compared to wild-type N2 worms (*Figure 4D*). Further evidence suggesting that PQM-1 activates *smf-3* was shown by reduced *smf-3* expression in *nhr-14(tm1473)* mutants fed *pqm-1* RNAi (*Figure 4E*) and by the low iron developmental delay observed in *pqm-1(ok485)* mutants (*Figure 4F*).

Given that PQM-1 transcriptionally activates *smf-3,* we questioned whether PQM-1 nuclear localization is affected by *nhr-14* loss. Previous studies showed that a PQM::GFP::FLAG transgene localized to the nucleus of intestinal cells during larval stages and to the cytoplasm in adults (*Dowen et al., 2016*; *O'Brien et al., 2018*; *Tepper et al., 2013*). In agreement with these studies, PQM-1::GFP::FLAG localized to nuclei in intestinal cells in L4 stage worms fed either control RNAi or *nhr-14* RNAi, and to the cytoplasm in adults fed control RNAi. PQM-1::GFP::FLAG, however, remained in intestinal nuclei in adult worms fed *nhr-14*-RNAi (*Figure 4G*). Together, these data

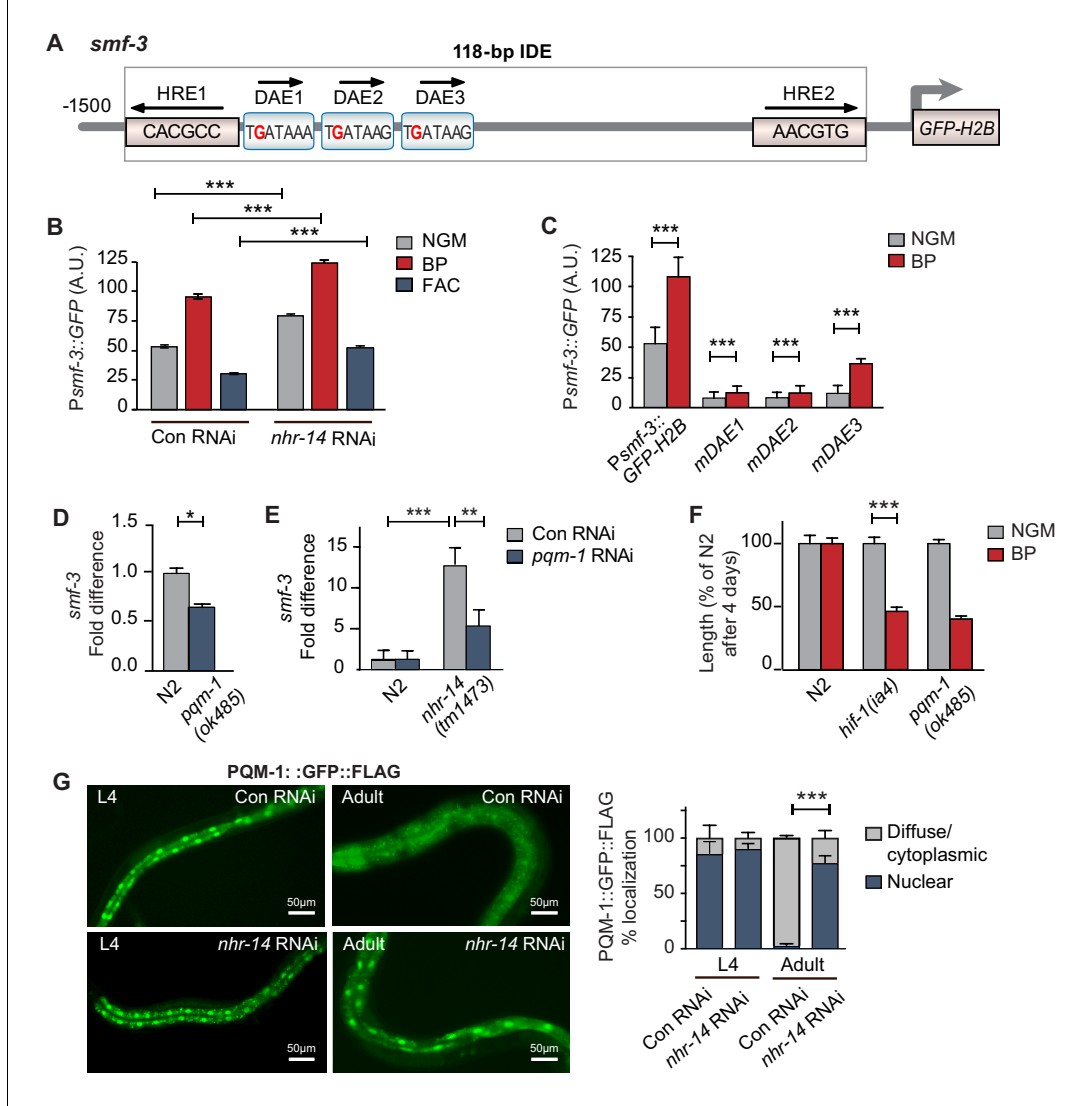

**Figure 4.** PQM-1 is downstream of NHR-14 and activates *smf-3*. (**A**) Illustration of the P*smf-3::GFP-H2B* transcriptional reporter containing 1.5 kb of 5'-promoter sequences of *smf-3*. The 118 bp iron-dependent element (IDE) harbors two hypoxia-response elements (HREs) and three GATA-like DAE motifs. Single nucleotide mutations in the DAEs are indicated in red. (**B**) GFP expression in P*smf-3::GFP-H2B* worms fed control (empty vector) RNAi or *nhr-14* RNAi grown on NGM, NGM-BP (low iron) or NGM-FAC (high iron) plates and quantified using COPAS Biosort (n = 1000 worms/sample). (**C**) GFP expression in P*smf-3::GFP-H2B* worms harboring a nucleotide mutation (G > C) in each DAE grown on NGM or NGM-BP plates and quantified using COPAS Biosort (n = 1000 worms/sample). (**D**) Endogenous *smf-3* expression in wild-type N2 worms versus *pqm-1(ok485)* mutants grown on NGM plates. (**E**) *smf-3* expression in wild-type N2 worms and *nhr-14(tm1473)* mutants fed control RNAi or *pqm-1* RNAi. (**F**) Length of wild-type N2, *hif-1(ia4)* and *pqm-1(ok485)* mutants grown on NGM or NGM-BP plates for 4 days. (**G**) PQM-1::GFP nuclear and cytoplasmic localization in L4 stage and adult worms fed control RNAi or *nhr-14* RNAi and quantification of PQM-1::GFP expression. L4 stage, control RNAi (n = 29) and *nhr-14* RNAi (n = 44); adults, control RNAi (n = 98) and *nhr-14* RNAi (n = 88). For B, E and G, values are expressed as mean ± SEM (n ≥ 3 biological replicates) and compared by two-way ANOVA with Tukey's multiple comparisons test, **$p < 0.01$, ***$p < 0.001$. For C, D and F, values are expressed as mean ± SEM (n ≥ 3 biological replicates) and are compared by unpaired two-tailed Student's *t* test, *$p < 0.05$, ***$p < 0.001$.

DOI: https://doi.org/10.7554/eLife.44674.008

The following figure supplements are available for figure 4:

**Figure supplement 1.** COPAS Biosort data for P*smf-3::GFP-H2B* transgene expression.
DOI: https://doi.org/10.7554/eLife.44674.009

**Figure supplement 2.** Validation of *nhr-14* RNAi potency.
DOI: https://doi.org/10.7554/eLife.44674.010

**Figure supplement 3.** PQM-1 shows significant association with the *smf-3* IDE.
DOI: https://doi.org/10.7554/eLife.44674.011

suggest that PQM-1 activates *smf-3* by associating with GATA-like DAE motifs in the *smf-3* promoter, and that NHR-14 acts upstream of PQM-1 to inhibit its nuclear translocation and transcriptional activity.

## NHR-14 controls expression of innate immune genes

In addition to regulation of intestinal iron uptake, RNA-seq analysis revealed that NHR-14 regulates genes involved in host defense (*Figure 5A–B*). We identified 573 upregulated genes and 261 downregulated genes in *nhr-14(tm1473)* mutants versus wild-type N2 worms using the filtering criteria of p-values<0.05 and fold change >1.0 (*Figure 3—source data 1*). Gene-set enrichment analysis using gene ontology (GO) terms (*Huang et al., 2009*) showed that upregulated genes were involved in the innate immune response (*Figure 5B*). Over-represented functional categories within this dataset included C-type lectin (carbohydrate binding proteins), CUB-domain proteins (extracellular and membrane proteins), lysosomes, peptidases, glutathione S-transferases (xenobiotic metabolism) and collagens, all of which are part of a core immune response program in *C. elegans* (*Simonsen et al., 2012*) (*Figure 5C*). Genes that are upregulated in *nhr-14(tm1473)* mutants include classical immune response genes, such as lysozymes (*lys-7, lys–1, lys-8, lys–2*), Class II downstream of *daf-16* (*dod-21, dod–24, dod-23, dod–19*), C-type lectins (*clec-41, clec-85, clec-190, clec–150, clec-173, clec–5, clec-63, clec–62*), antimicrobial peptides (*spp-18, spp-8*), and glutathione S-transferases (*gst-38, gst-22*) (*Figure 5D*). Using qPCR analysis, we verified *nhr-14*-dependent upregulation of ten genes within the immune response and iron metabolism groups (*Figure 5E*). In general, there was positive correlation between RNA-seq and qPCR expression values with the exception of *ftn-1*, which was significantly upregulated using qPCR, but not in the RNA-seq dataset. No significant change was observed in expression of iron/metal metabolism genes (*smf-1, smf-2, ftn-2*) and *pqm-1* in *nhr-14 (tm1473)* mutants (*Figure 5E*). Among the 261 downregulated genes, there was enrichment in genes involving cellular organization and body morphogenesis, including *sqt-1*, *noah-1* and *sym-1* (*Figure 5D* and *Figure 3—source data 1*). Analysis of tissue specific expression of the upregulated and downregulated *nhr-14* genes using WormExp v1.0 revealed enrichment in intestine and epidermis, respectively (*Figure 5—source data 1*).

An unbiased search for cis-regulatory elements in the 1 kb promotor regions of upregulated and downregulated *nhr-14* genes revealed the presence of a GATA motif in 341 out of 573 upregulated genes and a tandem-direct repeat motif in 14 out of 261 downregulated genes (*Figure 5F* and *Figure 3—source data 1*). The tandem-repeat is of interest as vertebrate HNF4α activates transcription by binding to DNA consisting of two distinct half-site motifs in either direct, inverted or everted configurations (*Fang et al., 2012*).

Given that the transcriptional profile of *nhr-14(tm1473)* mutants is enriched for immune response genes, we examined the overlap between upregulated *nhr-14* genes and previously published genes that are induced by infection to various pathogenic bacteria. We found overlap between genes upregulated in *nhr-14(tm1473)* mutants and genes upregulated in response to *P. aeruginosa* (*Shapira et al., 2006*; *Troemel et al., 2006*), *S. aureus* (*Visvikis et al., 2014*), *Pseudomonas luminescens* (*Wong et al., 2007*), *Enterococcus faecalis* (*Wong et al., 2007*) and *Serratia marcescens* (*Engelmann et al., 2011*) (*Figure 6A* and *Figure 6—source data 1*). These genes include lysozymes, C-type lectins, and antimicrobial proteins.

The pathogen response in *C. elegans* is controlled through three distinct signaling pathways that converge on inducing specific innate immune transcriptional program (*Cohen and Troemel, 2015*; *Irazoqui et al., 2010*). We compared our *nhr-14* RNA-seq dataset with p38-MAPK (*Troemel et al., 2006*), TGFβ (*Roberts et al., 2010*), ELT-2 (*Block et al., 2015*) and DAF-2/DAF-16 (*Murphy et al., 2003*) innate immune pathway datasets, and found expression overlap with these pathways (*Figure 6B* and *Figure 6—source datas 2* and *3*). Notably, overlap occurred with the DAF-16/FoxO-Class II subset of genes that include the downstream-of-*daf-16* (*dod*) group of which many are innate immune genes (*Murphy et al., 2003*) and classic antimicrobial genes (*Figure 6B* and *Figure 6—source data 3*. As Class II genes harbor GATA-like DAE motifs that are PQM-1 binding sites (*Tepper et al., 2013*), and GATA motifs are enriched in the promoters of upregulated *nhr-14* genes, this suggested that some of these genes may be regulated by PQM-1. Analysis of ChIP-seq data revealed significant PQM-1 association to 15 out of the top 25 upregulated genes in *nhr-14(tm1473)* mutants, while only 3 out of 25 for the downregulated subset showed significant PQM-1 association (*Niu et al., 2011*) (*Figure 6—source data 3*).

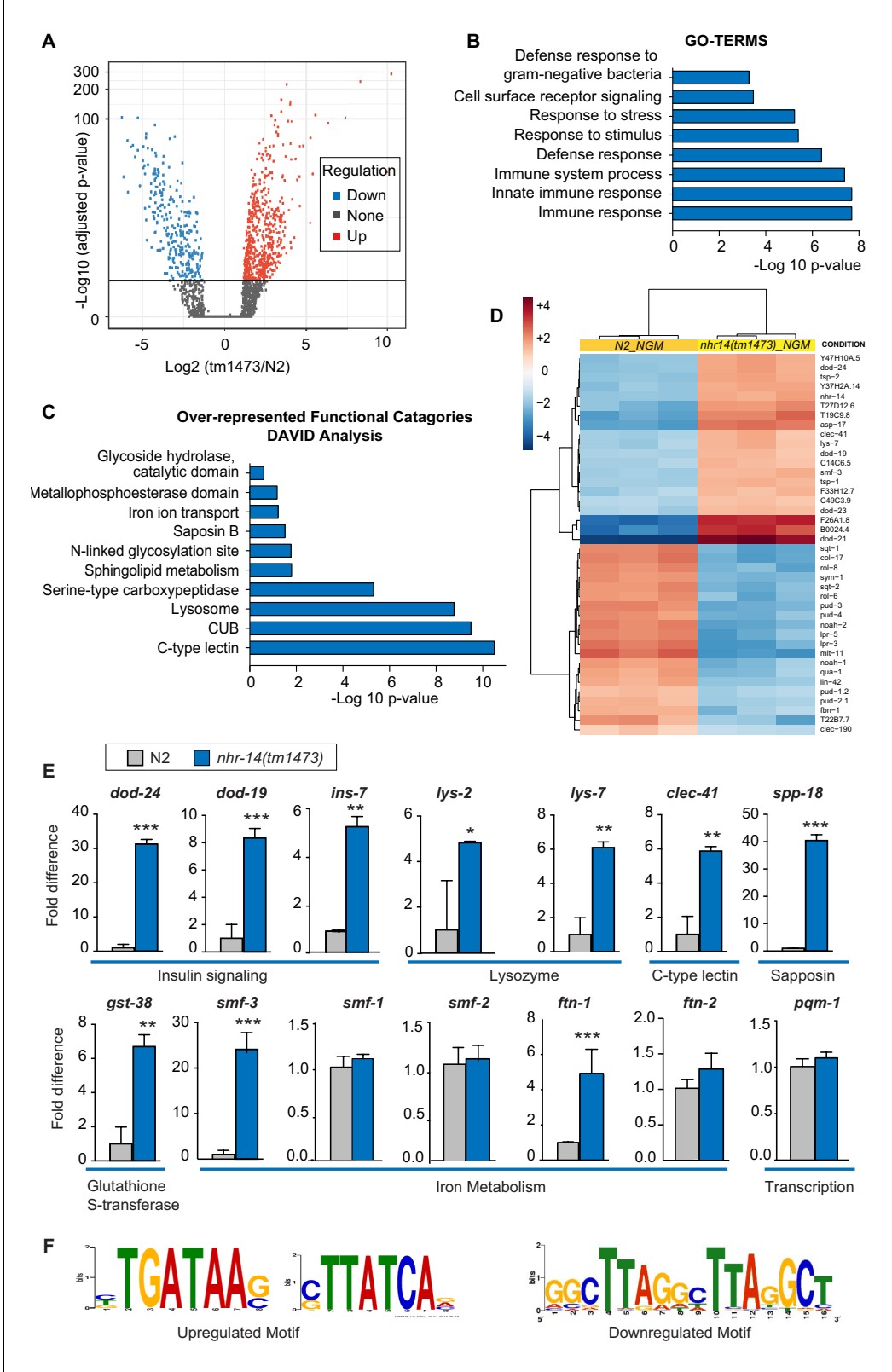

**Figure 5.** NHR-14 loss of function regulates an innate immune transcriptional program. (**A**) Volcano plot of differentially transcribed genes in *nhr-14 (tm1473)* mutants normalized to wild-type N2. The x-axis denotes the fold-changes and the y-axis denotes -log10 of wald test adjusted p-values. The horizontal line represents the *p*-value threshold of <0.05. (**B**) Gene ontology (GO) analysis of genes that are upregulated in *nhr-14(tm1473)* mutants normalized to wild-type N2 worms. (**C**) DAVID analysis for over-represented functional categories of upregulated genes in *nhr-14(tm1473)* mutants

*Figure 5 continued on next page*

*Figure 5 continued*

normalized to wild-type N2 worms. (**D**) Heat map showing the top 20 upregulated and 20 downregulated genes that are differentially expressed between *nhr-14(tm1473)* mutants and wild-type N2 worms. The columns represent three biological replicates of wild-type N2 and *nhr-14(tm1473)* mutants, and the rows represent individual genes. (**E**) qPCR validation of selected innate immune and iron-related genes that are upregulated in *nhr-14 (tm1473)* mutants (n ≥ 3 biological replicates). Values are expressed as fold difference compared to wild-type N2 worms ± SEM and compared by unpaired two-tailed Student's *t* test *p<0.05, **p<0.01, ***p<0.001. (**F**) Motif enrichment analysis for upregulated and downregulated genes in *nhr-14 (tm1473)* mutants.

DOI: https://doi.org/10.7554/eLife.44674.012

The following source data is available for figure 5:

**Source data 1.** Overlap between upregulated and downregulated *nhr-14(tm1473)* genes with published *C. elegans* tissue-specific expression datasets using WormExp v1.

DOI: https://doi.org/10.7554/eLife.44674.013

We next questioned whether other upregulated *nhr-14* genes in addition to *smf-3* require PQM-1 for expression. qPCR analysis showed reduced expression of Class II genes *dod-19*, *dod-24*, *clec-41*, *gst-38* and *oac-14* in *nhr-14(tm1473)*; *pqm-1* RNAi mutants compared with *nhr-14(tm1473)*; control RNAi worms. The Class II genes *ins-7* and *lys-2* and Class I genes *lys-7* and *ftn-1* were not changed in *nhr-14(tm1473)*; *pqm-1* RNAi mutants (*Figure 7*). Together, these data implicate NHR-14 as a

**A** Overlap between *nhr-14(tm1473)* mutants and genes induced upon infection with various pathogens

| Microarray data used in previous publications (reference) | Upregulated genes | Overlapping genes |
|---|---|---|
| *P. aeruginosa* (Shapira et al., 2006) | 196 | 62 |
| *P. aeruginosa* (Troemel et al., 2006) | 321 | 77 |
| *S. aureus* (Visvikis et al., 2014) | 821 | 47 |
| *P. luminescens* (Wong et al., 2007) | 660 | 60 |
| *E. faecalis* (Wong et al., 2007) | 641 | 46 |
| *S. marsescens* (Engelmann et al., 2011) | 2124 | 237 |

**B** Overlap between *nhr-14(tm1473)* mutants and published innate immune pathways and Class II genes

| Microarray data used in previous publications (reference) | Upregulated genes | Overlapping genes |
|---|---|---|
| *pmk-1*/p38 MAPK (Troemel et al.,2006) | 89 | 21 |
| *dbl-1*/TGFβ (Roberts et al., 2010) | 203 | 32 |
| *elt-2*/p38 MAPK (Block et al., 2015) | 127 | 36 |
| Class II DAF-16 (Murphy et al., 2003) | 250 | 29 |

**Figure 6.** Overlap between pathogen and innate immune pathway datasets and *nhr-14(tm1473)* mutants. (**A**) Number of overlapping genes between *nhr-14(tm1473)* mutants and published pathogen infection models. (**B**) Number of overlapping genes between *nhr-14(tm1473)* mutants and published innate immune pathway datasets.

DOI: https://doi.org/10.7554/eLife.44674.014

The following source data is available for figure 6:

**Source data 1.** Overlap between upregulated *nhr-14(tm1473)* genes at log fold change >1 and published pathogen infection dataset.

DOI: https://doi.org/10.7554/eLife.44674.015

**Source data 2.** Overlap between upregulated *nhr-14(tm1473)* genes at a log fold change >1 and published datasets for the p38/MAPK, TGFβ and *elt-2* pathways.

DOI: https://doi.org/10.7554/eLife.44674.016

**Source data 3.** Overlap between upregulated *nhr-14(tm1473)* genes at log fold change >1 and published datasets of DAF-16/FoxO-Class II genes, and PQM-1 ChIP-seq data.

DOI: https://doi.org/10.7554/eLife.44674.017

**Source data 4.** Overlap between upregulated and downregulated *nhr-14(tm1473)* genes with published *C. elegans* mutant-specific expression datasets using WormExp v1.

DOI: https://doi.org/10.7554/eLife.44674.018

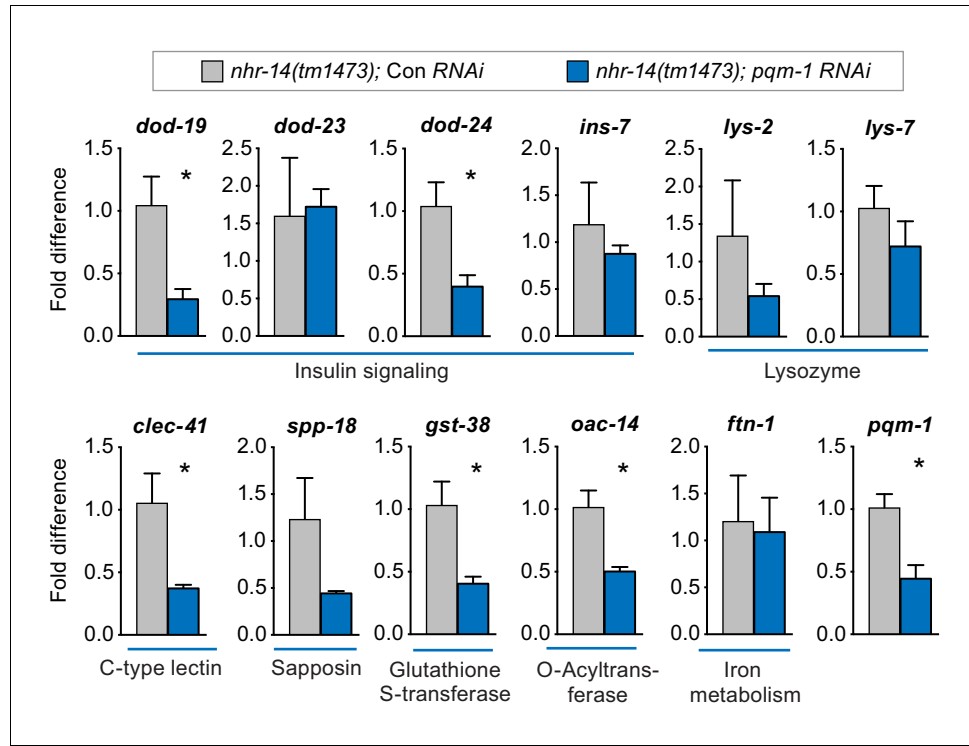

**Figure 7.** PQM-1 regulates DAF-16/FoxO-Class II genes downstream of *nhr-14*. qPCR analysis of upregulated *nhr-14* genes in *nhr-14(tm1473); pqm-1* RNAi mutants and *nhr-14(tm1473)*; Con RNAi worms (n = 3 biological replicates). Values are expressed as fold difference compared to *nhr-14(tm1473)*; Con RNAi mutants ± SEM and are compared by unpaired two-tailed Student's *t* test, *p<0.05.
DOI: https://doi.org/10.7554/eLife.44674.019

repressor of the transcriptional innate immune response and specific DAF-16/FoxO-Class II genes mediated in part by PQM-1.

Comparison of upregulated *nhr-14* genes with *C. elegans* mutant datasets using WormExp v1.0 showed significant overlap with genes upregulated in *nhr-8(hd117)* mutants and *hyl-2* mutants (*Figure 6—source data 4*). *nhr-8* encodes a nuclear receptor that regulates cholesterol and bile acid homeostasis (*Magner et al., 2013*) and *hyl-2* encodes ceramide synthase that is involved in oxygen deprivation and regulation of innate immune genes (*Ladage et al., 2016*). Downregulated *nhr-14* genes showed significant overlap with genes downregulated in *elo-5(gk208)* mutants (*Kniazeva et al., 2015*). *elo-5* encodes a fatty acid elongase that produces monomethyl branched-chain fatty acids that function in intestine to promote sensory neuron maturation (*Kniazeva et al., 2015*). These analyses suggest a role for NHR-14 in lipid metabolism.

## Loss of NHR-14 is required for host defense

Given that RNA-seq data revealed enrichment of innate immune response genes, DAF-16/FoxO-Class II suppressed genes and iron metabolism genes in *nhr-14(tm1473)* mutants, we tested the sensitivity of *nhr-14(tm1473)* and *smf-3(ok1035)* single mutants, and *smf-3(ok1035); nhr-14(tm1473)* double mutants to pathogenic *P. aeruginosa* strain PA14. PA14 causes a lethal intestinal infection in *C. elegans* similar to PA14 infection in mammals (*Kirienko et al., 2014*), and thus has become a useful model to study immune host defenses critical for resistance to pathogen infection (*Cornelis and Dingemans, 2013*). For our studies, the 'slow' killing assay was used that involves the colonization of *P. aeruginosa* in intestine (*Tan et al., 1999*). Compared to wild-type N2 worms, *nhr-14(tm1473)* mutants displayed enhanced resistance to PA14 infection, while *smf-3(ok1035)* mutants were hypersensitive (*Figure 8A* and *Figure 8—source data 1*). *smf-3(ok485); nhr-14(tm1473)* double mutants displayed increased resistance to PA14 compared to *smf-3(ok1035)* single mutants. These data show

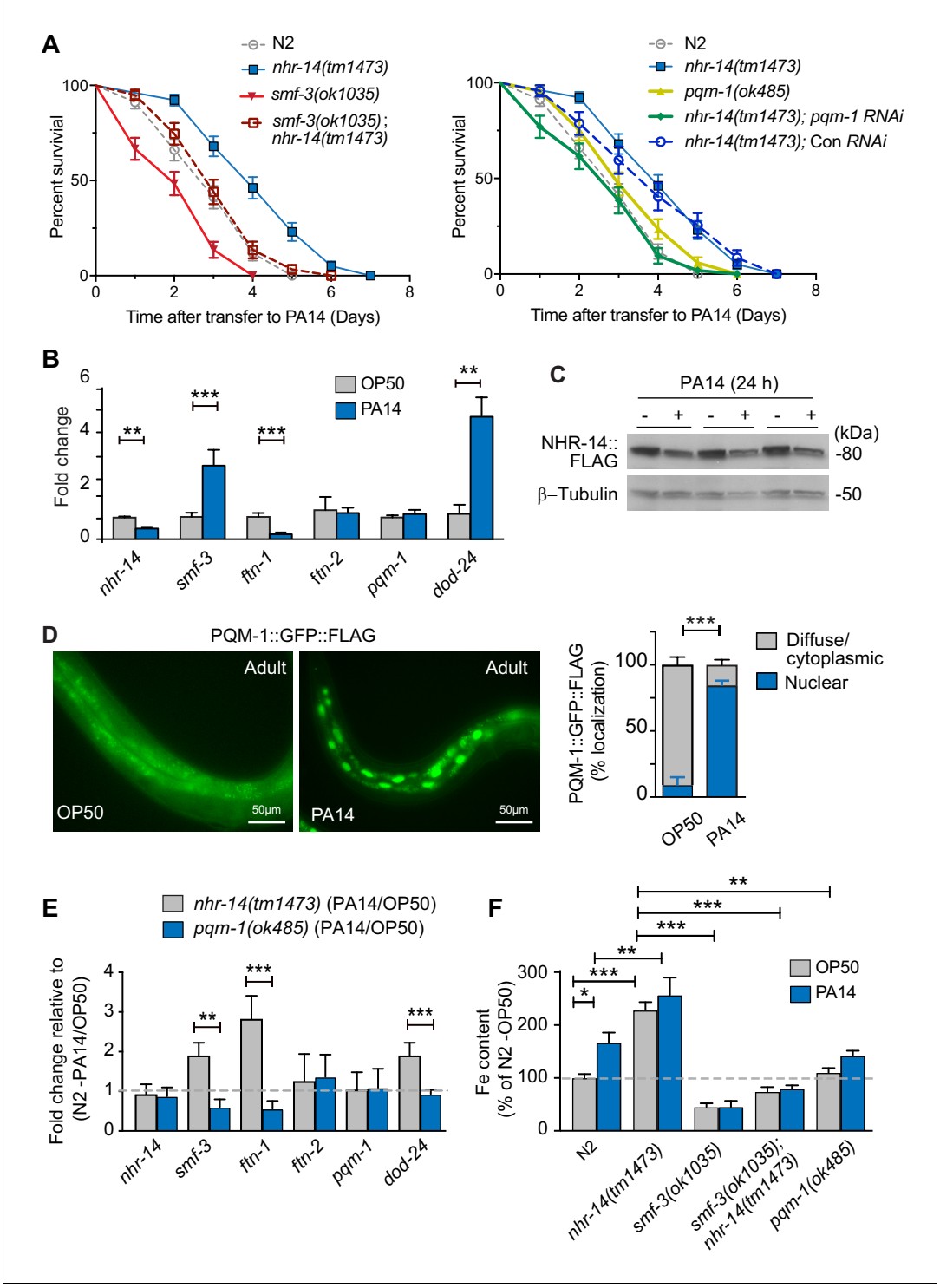

**Figure 8.** Loss of NHR-14 confers resistance to *P. aeruginosa* infection. (**A**) Survival analysis for the indicated mutants and wild-type N2 worms infected with *P. aeruginosa* PA14. Data are combined from three biological replicates. Median survival (MS): wild-type N2 (MS = 3 days, n = 68); *nhr-14(tm1473)* (MS = 4 days, n = 78); *pqm-1 (ok485)* (MS = 3 days, n = 68); *smf-3(ok1035)* (MS = 2 days, n = 66); *smf-3(ok1035); nhr-14(tm1473)* (MS = 3 days, n = 59); *nhr-14(tm1473);* Con (RNAi) (MS = 4 days, n = 47); *nhr-14(tm1473); pqm-1* (RNAi) (MS = 3 days, n = 52). Median survival of wild-type N2 worms grown in OP50 is 10 days (see *Figure 3D*). All survival curves are significantly different from one another, p<0.05, except for wild-type N2 vs *pqm-1(ok485)* p=0.835; wild-type N2 vs *smf-3(ok1035); nhr-14(tm1473)* p=0.621; *nhr-14(tm1473)* vs *nhr-14* Con RNAi; Log-rank Mantel-Cox test. (**B**) qPCR

*Figure 8 continued on next page*

*Figure 8 continued*
analysis of indicated genes in wild-type N2 worms after feeding on non-pathogenic *E. coli* OP50 or PA14 for 24 hr. Values are expressed as mean fold change relative to OP50 ± SEM (n = 3 biological replicates) and compared by unpaired two-tailed Student's *t* test, **p<0.01, ***p<0.001. (**C**) Western blot analysis of lysates prepared from NHR-14::GFP::FLAG worms after feeding on OP50 or PA14 for 24 hr (n = 3 biological replicates). (**D**) PQM-1::GFP:: FLAG expression in adult worms exposed to OP50 and PA14 for 24 hr. Quantification of nuclear and diffuse cytoplasmic localization of PQM-1::GFP:FLAG exposed to OP50 (n = 134 worms) and PA14 (n = 169 worms). Values are expressed as percentage PQM-1::GFP::FLAG localization. (**E**) qPCR analysis of indicated genes in *nhr-14(tm1473)* and *pqm-1(ok485)* mutants after feeding on OP50 or PA14 for 24 hr. Values are expressed as mean PA14/OP50 fold change relative to wild-type N2 worms ± SEM (n = 3 biological replicates) and compared by unpaired two-tailed Student's *t* test **p<0.01, ***p<0.001. (**F**) Iron content in wild-type N2 worms and indicated mutants after feeding on OP50 or PA14 for 24 hr as measured using ICP-MS. Values are expressed as mean ± SEM (n = 3–5 biological replicates) and compared by two-way ANOVA with Tukey's multiple comparisons test, *p<0.05, **p<0.01, ***p<0.001.

DOI: https://doi.org/10.7554/eLife.44674.020
The following source data is available for figure 8:

**Source data 1.** Survival statistics for *P. aeruginosa* strain PA14 lifespan analysis.
DOI: https://doi.org/10.7554/eLife.44674.021

that *nhr-14* partially suppresses the sensitivity of *smf-3(ok1035)* to PA14, suggesting that other genes in addition to *smf-3* contribute to the resistance of *nhr-14(tm1473)* mutants to PA14.

We also analyzed the resistance of the *pqm-1(ok485)* single mutant and *nhr-14(tm1473); pqm-1* RNAi mutants to PA14. *pqm-1(ok485)* mutants displayed a wild-type sensitivity to PA14. *nhr-14 (tm1473); pqm-1* RNAi mutants were more sensitive to PA14 than *nhr-14(tm1473)* single mutants, indicating that PQM-1 is required for *nhr-14(tm1473)* enhanced PA14 resistance (*Figure 8A* and *Figure 8—source data 1*).

We tested whether PA14 affects the expression of *nhr-14, smf-3, ftn-1* and *ftn-2* in wild-type N2 worms. The DAF-16/FoxO-Class II *dod-24* gene was also examined as it is known to be induced by PA14 (*Shapira et al., 2006*; *Troemel et al., 2006*). PA14 exposure (24 hr) repressed *nhr-14* and *ftn-1* and induced *smf-3* and *dod-24,* but did not affect *ftn-2* or *pqm-1* compared to non-pathogenic OP50 *E. coli* (*Figure 8B*). PA14-mediated suppression of *nhr-14* mRNA correlated with reduced NHR-14::GFP::FLAG expression in transgenic worms exposed to PA14 for 24 hr (*Figure 8C*). We also found that PQM-1::GFP::FLAG accumulated in the nucleus upon PA14 exposure in adult worms (*Figure 8D*), consistent with PQM-1 nuclear localization in adult worms fed *nhr-14* RNAi (*Figure 4G*).

We next questioned whether *nhr-14* or *pqm-1* is required for expression of *smf-3, ftn-1, ftn-2* and *dod-24* during a PA14 infection. For these experiments, we assessed the expression of the above genes in *nhr-14(tm1473)* and in *pqm-1(ok485)* mutants after exposure to OP50 and PA14 for 24 hr. PA14 induced expression of *smf-3, dod-24* and *ftn-1* in *nhr-14(tm1473)* mutants relative to wild-type N2-PA14/OP50 levels, and notably, these genes were repressed by PA14 in *pqm-(ok485)* mutants (*Figure 8E*). *ftn-2* was unchanged in *nhr-14(tm1473)* and *pqm-1(ok485)* mutants, and *pqm-1* was unchanged in *nhr-14(tm1473)* mutants exposed to either OP50 or PA14 (*Figure 8F*). These data show that NHR-14 repression and PQM-1 are required for PA14 induction of *smf-3, ftn-1* and *dod-24*.

Iron is required for pathogen survival and infectivity, and both pathogens and hosts have evolved specialized mechanisms to control iron availability (*Hood and Skaar, 2012*). We postulated that *P. aeruginosa*-mediated NHR-14 repression may provide a strategy to limit iron to the pathogen in the intestinal lumen by increasing iron uptake by SMF-3. If so, iron content would increase in wild-type N2 worms exposed to PA14. As predicted, iron content increased in wild-type N2 worms exposed to PA14 compared to OP50, and was further increased in *nhr-14(tm1473)* mutants exposed to either OP50 or PA14 (*Figure 8F*). The low iron content observed in *smf-3(ok1035)* single mutants and *smf-3(ok1035); nhr-14(tm1473)* double mutants was unaffected by PA14 exposure (*Figure 8E* and *Figure 3B*). Iron content was lower in *pqm-1(ok485)* mutants than *nhr-14(tm473)* mutants, and was unaffected by PA14. Together, these data show that NHR-14 repression and PQM-1 activation are required for PA14 induction of SMF-3-dependent iron uptake and FTN-1 iron storage, demonstrating iron sequestration as a component of the innate immune response in *C. elegans* (*Figure 9*).

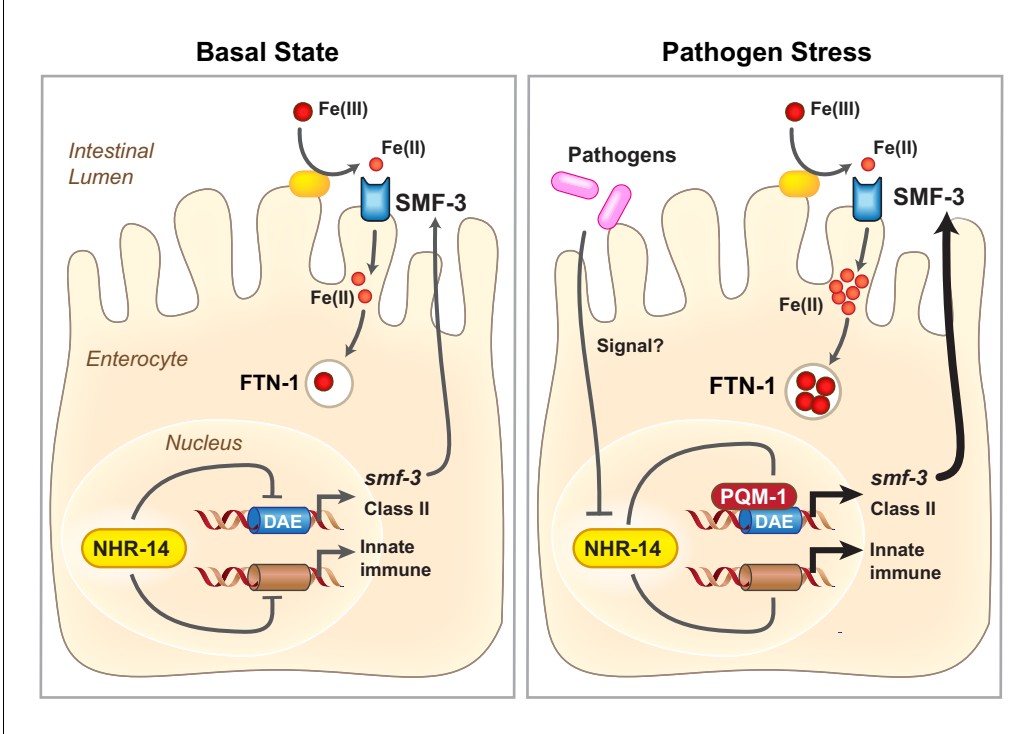

**Figure 9.** Proposed model for NHR-14-PQM-1 signaling pathway in intestinal regulation of innate immunity and iron metabolism in *C. elegans*. Under basal conditions, NHR-14 represses *smf-3*, specific DAF-16/FoxO-Class II genes and innate immune response genes to promote growth and development. Upon exposure to *P. aeruginosa*, NHR-14 function/activity is reduced, leading to PQM-1 nuclear localization and transcriptional activation of *smf-3* and DAF-16/FoxO-Class II through the GATA-like DAE. SMF-3 mediates Fe(II) uptake after the reduction of Fe (III) by an unknown membrane associated ferrireductase (yellow oval). Increased cellular iron activates *ftn-1* transcription through a HIF-1-dependent mechanism (*Ackerman and Gems, 2012*; *Romney et al., 2011*), leading to the sequestration of iron within FTN-1. We suggest that increased iron uptake by SMF-3 and sequestration by FTN-1 may provide a strategy to limit *P. aeruginosa* acquisition of intestinal luminal iron and serves as a component of the host innate immune response. How NHR-14 senses pathogen stress and the identity of NHR-14 downstream effectors remain to be determined.
DOI: https://doi.org/10.7554/eLife.44674.022

## Discussion

Using a genetic suppressor screen to rescue the low iron developmental delay phenotype of *hif-1 (ia4)* mutants, we identified NHR-14 as a transcriptional repressor of SMF-3-dependent iron uptake and innate immunity in *C. elegans*. Loss of *nhr-14* increases SMF-3-mediated iron uptake and induces innate immune response genes that contribute to the resistance of *nhr-14(tm1473)* mutants to *P. aeruginosa*. These transcriptional responses are regulated in part by the transcription factor PQM-1. Our data support a model whereby *C. elegans* respond to pathogen infection by reducing NHR-14 abundance, thus allowing PQM-1 to translocate to the nucleus of intestinal cells to activate genes critical for innate immunity and iron uptake. Several *C. elegans* nuclear receptors, such as DAF-12 (*Liu et al., 2013*), NHR-49 (*Sim and Hibberd, 2016*), NHR-25 (*Ward et al., 2014*), and NHR-114 and NHR-144 (*Yuen and Ausubel, 2018*), regulate innate immune response genes; however, there is no evidence to date indicating a role for these nuclear receptors in iron metabolism. NHR-14 may thus be unique among nuclear receptors in coupling innate immunity with iron sequestration as a strategy to limit iron to invading intestinal pathogens.

Originally discovered to be induced in response to paraquat-mediated stress (*Tawe et al., 1998*), PQM-1 has been shown to be responsible for the activation of the DAF-16/FoxO-Class II subset of IIS genes (*Tepper et al., 2013*). Under normal conditions, the IIS pathway is upregulated by insulin-like peptides via activation of the DAF-2 tyrosine kinase receptor that culminates in the transcription

factor DAF-16/FoxO phosphorylation and its cytoplasmic retention. Under stress, IIS signaling is reduced, DAF-16/FoxO is dephosphorylated and driven into the nucleus to activate Class I genes (stress response) and repress Class II genes (metabolism/growth) (*Murphy et al., 2003*; *Tepper et al., 2013*). PQM-1 antagonizes DAF-16/FoxO nuclear localization upon increased insulin signaling, resulting in the activation of Class II genes through binding to the DAE (*Tepper et al., 2013*). In larval stage worms, PQM-1 localizes to the nucleus of intestinal cells, but in adults, PQM-1 becomes mostly cytoplasmic (*Dowen et al., 2016*; *O'Brien et al., 2018*; *Tepper et al., 2013*). Consistent with these studies, we found PQM-1::GFP localized to the nucleus in L4 stage worms fed either control RNAi or *nhr-14* RNAi; however, PQM-1::GFP remained nuclear in adults fed *nhr-14* RNAi. These data suggest that PQM-1 nuclear localization caused by loss of *nhr-14* in adults may be a mechanism to maintain growth via Class II gene activation as well as defend against stress in aged worms. Other signaling pathways are known to regulate PQM-1 subcellular localization, including mTORC/SGK-1, which promotes PQM-1 cytoplasmic localization in intestinal cells (*Dowen et al., 2016*), and HSP-90, which promotes PQM-1 nuclear localization in neuronal and intestinal cells (*O'Brien et al., 2018*). PQM-1 has also been shown to have a role in reproductive aging (*Templeman et al., 2018*).

The enrichment of Class II and innate immune response genes in *nhr-14(tm1473)* mutants suggest that these mutants may have a survival advantage when challenged with pathogens. As predicted, *nhr-14(tm1473)* mutants displayed enhanced resistance to *P. aeruginosa* that require PQM-1. *P. aeruginosa* infection reduced NHR-14 abundance in wild-type N2 adults, which led to PQM-1 nuclear localization, activation of *smf-3* and specific DAF-16/FoxO-Class II genes and increased iron content. The observation that not all DAF-16/FoxO-Class II genes are regulated by PQM-1 in *nhr-14(tm1473)* mutants suggests other pathways are involved in their regulation. Our data are consistent with a model whereby increased iron uptake by SMF-3 may be a strategy to limit iron to pathogens in the intestinal lumen. It is also possible that iron helps to maintain iron homeostasis during a pathogen infection as it is required by enzymes involved in critical cellular processes.

We found that *pqm-1(ok485)* mutants displayed similar sensitivity to *P. aeruginosa* PA14 as wild-type N2 worms consistent with a recent report (*O'Brien et al., 2018*), but conflicts with two other studies showing *pqm-1* is required for resistance to PA14 (*Joshi et al., 2016*; *Shapira et al., 2006*). The reason for the conflicting data is unclear, but may be caused by different background strains or different conditions used for pathogen survival assays.

In vertebrates, natural resistance-associated macrophage protein 1 (NRAMP1), a paralog of DMT1, is found in the phagosome membrane of macrophages, where it transports Fe(II) and Mn(II) out of phagosomes, thus restricting growth of pathogens residing within these compartments by limiting iron availability (*Lopez and Skaar, 2018*; *Wessling-Resnick, 2015*). DMT1 is highly expressed in duodenal enterocytes where it is a key regulator of intestinal iron absorption (*Fleming et al., 1998*; *Gunshin et al., 1997*). The intestinal mucosa also serves as the first line of defense against pathogens. Upon a pathogen infection in intestine, the innate immune response is triggered by the detection of microorganism-associated molecular patterns (MAMPs) via pattern-recognition receptors (PRR) (e.g. Tol/Toll-like and NOD-like receptors) that activate pathways regulating the production of anti-microbial proteins and cytokines, as well as the recruitment of acute inflammatory cells (*Lopez and Skaar, 2018*; *Santaolalla et al., 2011*). Several studies have shown that DMT1 is induced in response to TNFα (tumor necrosis factor α) and IFNγ (interferon γ) in duodenal enterocytes and bronchial epithelial cells (*Laftah et al., 2006*; *Wang et al., 2005*). It was speculated in these studies that the increase in DMT1 is a means to sequester iron to prevent microbial access to host iron and limit pathogenesis. Further investigation is needed to determine a role for DMT1-mediated iron uptake in intestinal innate immunity in vertebrates.

How NHR-14 senses pathogens remain to be determined. Unlike vertebrates, *C. elegans* lack clear pattern-recognition receptors orthologs, and its single Toll-like receptor is not necessary for immune defense (*Cohen and Troemel, 2015*; *Stuart et al., 2013*). A surveillance pathway has been identified in *C. elegans* that detects microbial-derived toxin damage to core cellular processes, such as translation and mitochondrial respiration, and induces signaling pathways controlling innate immune genes (*Pukkila-Worley, 2016*). The ligand-binding domain of NHR-14 may sense and bind pathogen-derived molecules or endogenous-produced ligands that facilitate NHR-14 stability or coregulator binding to regulate target genes. NHR-14 is also highly expressed in head neurons, suggesting that neuronal NHR-14 may detect small molecules that could modulate the innate immune

response in intestine. NHR-14 was reported to function as an estrogen receptor (*Mimoto et al., 2007*), but conflicts with other studies showing that estrogen and bisphenol A did not affect NHR-14 function (*Allard and Colaiácovo, 2010*; *Fischer et al., 2012*).

Of the 284 NRs in *C. elegans*, 269 are derived from a HNF4-related ancestor (*Robinson-Rechavi et al., 2005*). This is unlike mammals that express two HNF4 receptors (HNF4α and HNF4γ) and *Drosophila melanogaster* that express a single HNF4 (*Taubert et al., 2011*). Mammalian HNF4α is expressed in liver, kidney, intestine, and the colon where it activates genes involved in glucose, fatty acid, and xenobiotic metabolism (*Gonzalez, 2008*). HNF4α has been shown to control iron homeostasis in liver and hepatoma cells through the regulation of transferrin, transferrin receptor-1 and hepcidin genes (*Matsuo et al., 2015*; *Truksa et al., 2009*). The expansion of the HNF4-related NRs in *C. elegans* suggests that they may have evolved dedicated functions that are performed by HNF4 in higher organisms (*Taubert et al., 2011*). Studies have shown that in the plant parasitic nematode *Meloidogyne incognita* (*Abad et al., 2008*) and in the human parasitic nematode *Brugia malayi* (*Ghedin et al., 2007*) the number of NRs are reduced to 92 and 27, respectively. Of note, NHR-14 is one of four HNF4-related nuclear receptors that is conserved in both parasites, suggesting that NHR-14 may be important for parasite infectivity.

In summary, the identification of NHR-14 adds a new layer of regulation that links iron metabolism with the innate immune response and provides new insights by which *C. elegans* responds to pathogen infections.

# Materials and methods

### Key resources table

| Reagent type (species) or resource | Designation | Source or reference | Identifiers | Additional information |
|---|---|---|---|---|
| Strain (*Caenorhabditis elegans*) | *C. elegans* strains used in this study are listed in *Supplementary file 1* - Table 1 | This paper | | |
| Strain (*Escherichia coli*) | OP50 | *C. elegans* Genetics Center | RRID: WB-STRAIN:OP50 | |
| Strain (*Escherichia coli*) | HT115(DE3) | *C. elegans* Genetics Center | RRID: WB-STRAIN:HT115(DE3) | |
| Strain (*Pseudomonas aeruginosa*) | PA14 | Other | RRID: WB-STRAIN:PA14 | Ausubel Laboratory, The Massachusetts General Hospital |
| Genetic reagent | *nhr-14* RNAi (T01B10) | *Kamath et al., 2003*; PMID: 12828945 | Ahringer RNAi feeding library | |
| Genetic reagent | *pqm-1* RNAi (F40F8.7) | *Kamath et al., 2003*; PMID: 12828945 | Ahringer RNAi feeding library | |
| Antibody | Monoclonal anti-FLAG M2 antibody | Sigma Aldrich | Cat# F3165; RRID: AB_259529 | |
| Antibody | Monoclonal anti β-tubulin antibody | MP Biomedicals LLC | Cat# 08691261; RRID: AB_2335131 | |
| Sequence-based reagent - TaqMan Assay | *smf-3* | Thermofisher Scientific | Ce02461546_g1 | |
| Sequence-based reagent - TaqMan Assay | *spp-18* | Thermofisher Scientific | Ce02457997_g1 | |
| Sequence-based reagent - TaqMan Assay | *nhr-14* | Thermofisher Scientific | Ce02420450_g1 | |

*Continued on next page*

*Continued*

| Reagent type (species) or resource | Designation | Source or reference | Identifiers | Additional information |
|---|---|---|---|---|
| Sequence-based reagent - TaqMan Assay | *act-2* | Thermofisher Scientific | Ce02507510_s1 | |
| Sequence-based reagent - TaqMan Assay | *dod-19* | Thermofisher Scientific | Ce02470201_m1 | |
| Sequence-based reagent - TaqMan Assay | *dod-24* | Thermofisher Scientific | Ce02466435_g1 | |
| Sequence-based reagent - TaqMan Assay | *lys-2* | Thermofisher Scientific | Ce02480494_g1 | |
| Sequence-based reagent - TaqMan Assay | *lys-7* | Thermofisher Scientific | Ce02473444_g1 | |
| Sequence-based reagent - TaqMan Assay | *clec-41* | Thermofisher Scientific | Ce02483609_g1 | |
| Sequence-based reagent - TaqMan Assay | *gst-38* | Thermofisher Scientific | Ce02486571_g1 | |
| Sequence-based reagent - TaqMan Assay | *ftn-1* | Thermofisher Scientific | Ce02477612_g1 | |
| Sequence-based reagent - TaqMan Assay | *ftn-2* | Thermofisher Scientific | Ce02415799_g1 | |
| Sequence-based reagent - TaqMan Assay | *smf-1* | Thermofisher Scientific | Ce02496641_g1 | |
| Sequence-based reagent - TaqMan Assay | *smf-2* | Thermofisher Scientific | Ce02496629_g1 | |
| Sequence-based reagent - TaqMan Assay | *nhr-23* | Thermofisher Scientific | Ce02405513_g1 | |
| Sequence-based reagent - TaqMan Assay | *pqm-1* | Thermofisher Scientific | Ce02438957_g1 | |
| Sequence-based reagent - TaqMan Assay | *tba-1* | Thermofisher Scientific | Ce02412618_gH | |
| Sequence-based reagent - TaqMan Assay | *ins-7* | Thermofisher Scientific | Ce02458078_g1 | |
| Sequence-based reagent - TaqMan Assay | *dod-23* | Thermofisher Scientific | Ce02435949_g1 | |
| Sequence-based reagent - TaqMan Assay | *oac-14* | Thermofisher Scientific | Ce02499634_g1 | |
| Recombinant DNA | L440 plasmid | Addgene | Plasmid # 1654; https://www.addgene.org/1654/ RRID: Addgene_1654 | |

*Continued on next page*

*Continued*

| Reagent type (species) or resource | Designation | Source or reference | Identifiers | Additional information |
|---|---|---|---|---|
| Recombinant DNA | NHR-14::3xFLAG::GFP transgene fosmid | *Sarov et al., 2006*; PMID: 22901814, TransgeneOME | WBGene0000361 | |
| Chemical compound, drug | 2,2'-Bipyridyl | Sigma Aldrich | Cat# D216305 | |
| Chemical compound, drug | Ferric Ammonium Citrate | Sigma Aldrich | Cat# RES20400-A7 | |
| Chemical compound, drug | 5-fluoro-2-deoxyuridine (FUdR) | Sigma Aldrich | Cat# F0503 | |
| Chemical compound, drug | Coomassie Plus Protein Assay Reagent | Thermofisher Scientific | Cat# 23236 | |
| Chemical compound, drug | Isopropyl β-D-1-thiogalactopyranoside (IPTG) | Thermofisher Scientific | Cat# 15529019 | |
| Chemical compound, drug | TRIzol Reagent | Invitrogen | Cat# 15596018 | |
| Software, algorithm | Prism 6 and 7.04 | Graphpad | https://www.graphpad.com/ RRID: SCR_002798 | |
| Software, algorithm | Image J | National Institute of Health | https://imagej.nih.gov/ij/ | |
| Software, algorithm | Excel 2013 | Microsoft Professional 2013 Spreadsheet Software | https://products.office.com/en-us/excel | |
| Software, algorithm | BioProspector | *Liu et al., 2001* | http://ai.stanford.edu/~xsliu/BioProspector/ | |
| Software, algorithm | DAVID Bioinformatics Resources 6.8 | *Huang et al., 2009*; PMID: 19131956 | https://david.ncifcrf.gov/home.jsp | |
| Commercial assay or kit | Western Lighting Plus-ECL Chemiluminescence | Perkin Elmer | Cat# WP20005 | |
| Commercial assay or kit | qScript XLT cDNA SuperMix | QuantaBio | Cat# 95161–025 | |
| Commercial assay or kit | SuperScript III First-Strand Synthesis SuperMix | Invitrogen | Cat# 18080400 | |
| Commercial assay or kit | Pierce Coomassie Plus (Bradford) Assay Reagent | Thermofisher Scientific | Cat# 23238 | |
| Commerical assay or kit | Pierce BCA Protein Assay | Thermofisher Scientific | Cat# 23227 | |

## *C. elegans* strains and culture

Strains obtained from the CGC and those generated for this study are listed in *Supplementary file 1* and Key Resources Table. *nhr-14(tm1473)* mutants were backcrossed 6x to wild-type N2 worms. These worms displayed normal development and brood size. *hif-1(ia4); nhr-14(tm1473)* and *smf-3 (ok1035); nhr-14(tm1473)* double mutants, and *smf-3(ok1035); hif-1(ia4); nhr-14(tm1473)* triple mutants were constructed and verified using standard genetic and molecular methods (*Brenner, 1974*). Worms were grown on nematode growth medium (NGM) agar plates seeded with *Escherichia coli* OP50 bacteria at 22°C unless otherwise noted. For iron experiments, worms synchronized by hypochlorite treatment at the L4 stage to young adult stage were grown overnight on NGM plates and then transferred to NGM plates supplemented with 6.6 mg/ml ferric ammonium citrate (FAC) or 25 uM 2,2'-bipyridyl (BP, iron chelator) for rescue experiments and 100 uM BP for

RNAi experiments for times indicated in each experiment. The pH of FAC-NGM agar was adjusted to pH 7.0. For synchronizing worms, eggs were obtained by treating gravid adults with alkaline hypochlorite. Eggs were allowed to hatch overnight in Egg Buffer (118 mM NaCl, 48 mM KCl, 2 mM CaCl$_2$, 2 mM MgCl$_2$, and 25 mM HEPES) and arrest in the L1 stage. Synchronized L1 stage worms were grown on NGM plates seeded with OP50 until reaching the L4 stage.

## Reporter constructs

P*smf-3::GFP-H2B* was generated by PCR amplification of sequences 1,500 bp upstream from the initiation ATG of *smf-3* (Y69A2AR.4) using primers containing *SalI* and *NheI* restrictions sites, cloned into TOPO Zeroblunt (Invitrogen) followed by digestion and insertion into *Sal1* and *NheI* sites of pAP.10 as previously described (*Romney et al., 2011*). The GATA-like DAE 1–3 mutants were generated by mutation of <u>G</u>ATA ><u>C</u>ATA in P*smf-3::GFP-H2B* using the QuikChange Site-Directed Mutagenesis kit (Stratagene). Transgenic strains were generated using a standard microinjection protocol. Each DAE construct (20 ng/µl), a selection plasmid pBX-1 (100 ng/µl) and salmon sperm DNA (25 ng/ul) were coinjected into *pha-1* animals. Transgenic animals were obtained after growth at 25°C. Five independent lines were established for P*smf-3(DAE1)::GFP-H2B*, P*smf-3(DAE2)::GFP-H2B* and P*smf-3(DAE3)::GFP-H2B*, and all showed similar GFP patterns.

## Generation of NHR-14::FLAG::GFP transgenic worms

The expression pattern of NHR-14 was determined using transgenic animals expressing a fosmid (40 kb) encoding a NHR-14::3xFLAG::GFP transgene (TransgeneOme) (*Sarov et al., 2006*). This fosmid contains the entire *nhr-14* locus as well as large amounts of flanking DNA critical for endogenous *nhr-14* expression. The fosmid was injected into *ttTi5606; unc-119* worms at 250 ng/ul and three independent lines were established. Line three was used for experiments in *Figure 2A-D*.

## EMS screen and SNP mapping

Ethyl methanesulfonate (EMS) mutagenesis was performed as previously described (*Brenner, 1974*). Briefly, L4 stage *hif-1(ia4)* worms (*Jiang et al., 2001*) were treated with 50 mM EMS in M9 buffer (22 mM KH$_2$PO$_4$, 42 mM Na$_2$HPO$_4$, 86 mM NaCl) for 4 hr. Mutagenized worms were washed five times to remove excess EMS, plated on NGM plates and grown until gravid. Gravid F1s were then bleached, and F2 eggs plated on NGM plates. Gravid F2 worms were transferred to 25 uM BP plates and removed after laying F3 eggs. F3 progeny that rescued the low iron phenotype of *hif-1(ia4)* mutants (determined by body length measurements using ImageJ) were scored as suppressors. We identified 16 suppressors from 15,000 haploid genomes using EMS, six that fully and 10 that partially rescued the *hif-1(ia4)* low iron phenotype. We focused on two mutations, *qa6909* and *qa6910*, that strongly suppressed the *hif-1(ia4)* low iron phenotype. The other four strong suppressors were sequenced and were not mutations in *nhr-14*.

Whole genome sequencing-single nucleotide polymorphism (WGS-SNP) mapping was used to identify the causative mutations in *qa6909* and *qa6910* (*Doitsidou et al., 2010*). *hif-1(ia4)* mutants were crossed into the polymorphic Hawaiian strain CB4856 six times. Hawaiian *hif-1(ia4)* males were crossed with *qa6909* and *qa6910* mutants and F1 progeny were moved to fresh NGM plates. Gravid F1 adults were allowed to lay eggs overnight on NGM plates containing 25 uM BP. Adults were removed the next day and ~50 F2 recombinants that reached adulthood within five days were singled to NGM plates and DNA from their F3 and F4 progeny was purified using the PureLink Genomic DNA Mini kit (Invitrogen) and pooled. WGS was performed using an Illumina HiSeq sequencer. Hawaiian SNP positions were mapped on the genomes of suppressor mutants *qa6909* and *qa6910* using the single-step SNP-mapping pipeline CloudMap (*Minevich et al., 2012*).

## Rescue experiments

Five gravid adults from each strain were picked to either NGM or NGM-25 uM BP plates and allowed to lay eggs overnight. Adults were removed the following day and progeny were incubated for five days at 22°C after which images were collected using a Leica M205 FA microscope and Leica DFC 310 FX camera (Leica Microsystems). Progeny were scored based on length using Image J.

## RNAi

RNAi clones *nhr-14* (T01B10) and *pqm-1* (F40F8.7) were obtained from the Ahringer RNAi feeding library (*Kamath et al., 2003*). Empty vector (L4440) was used as a control. All RNAi clones were verified by sequencing. Bacterial strains (HT115 DE3) were grown overnight in Luria-Bertani medium containing 50 µg/ml ampicillin and seeded onto NGM or NGM-BP plates containing 100 µg/ml ampicillin and 1 mM isopropyl β-D-1-thiogalactopyranoside (IPTG) to induce the gene of interest. Bacteria were then incubated at room temperature overnight to induce dsRNA. Five *hif-1(ia4)* gravid adults were allowed to lay eggs overnight on NGM or NGM-25 uM BP RNAi plates at 20˚C. Adults were removed the following day and progeny were scored for rescue against BP as described above. For qPCR experiments, 20–30 gravid adults were placed on RNAi plates for 48–72 hr. The progeny that reached L3 and L4 stages were harvested for RNA isolation.

For *nhr-14(tm1473); pqm-1* RNAi experiments, L1 stage synchronized worms were seeded onto RNAi plates and incubated at 22˚C to obtain adult hermaphrodites (P0). Gravid hermaphrodites from P0 were used to obtain a L1 stage synchronized population (F1), which was seeded onto *pqm-1* RNAi plates for a second round. Gravid F1 hermaphrodites were synchronized to obtain a L1 stage population (F2) and were seeded on *pqm-1* RNAi plates for a third round. Worms were allowed to reach L4 stage and then harvested for qPCR experiments.

## qPCR

For indicated conditions and time-points, worms were washed off the plate with M9 buffer followed by three further washes to remove external bacteria, and rocked for 0.5 hr in M9 buffer to clear bacteria from the gut. RNA was extracted using TRIzol Reagent (Invitrogen) according to the manufacturer's protocol. cDNA synthesis was performed using SuperScript III First-Strand Synthesis SuperMix (Invitrogen) or qScript XLT cDNA SuperMix (QuantaBio). After reverse transcription, qPCR was performed using TaqMan Gene Expression Assays (Applied Biosystems) and analyzed using an Applied Biosystems 7900 HT qPCR instrument. The cycle threshold (Ct) value for each transcript was normalized to either *act-1*, *act-2b* or *nhr-23*. For pathogen experiments, all values were normalized using control genes *nhr-23* or *tba-1*. The comparative Ct method was used to quantify transcript abundance. At least three biological replicates with duplicate technical replicates were used for qPCR experiments. TaqMan Assays are listed in Key Resources Table.

## RNA-seq analysis

Wild-type N2 and *nhr-14(tm1473)* L1 stage worms were grown on NGM plates seeded with OP50 and harvested at L4 stage (48 hr at 22˚C). Three biological replicates for each strain were used for RNA-seq analysis. Worms were collected by centrifugation and washed with M9 buffer to remove, and RNA was extracted using a modified TRIzol procedure. Briefly, the aqueous phase was mixed with an equal volume of 70% EtOH and added to RNeasy spin columns (Qiagen). RNA concentration and quality were determined using a NanoDrop spectrophotometer followed by further quality control with the Bioanalyzer RNA 9000 Nano Chip. Library construction was performed using the Illumina TruSeq RNA Sample Preparation Kit v2 (RS-122–2001 and RS-122–2002) and was sequenced with the Illumina HiSeq (50 cycle, single-end). Sequenced reads were aligned to the Ce10 (WormBase Build 220) transcriptome index using Novoalign, and differential expression was determined using Useq 8.7.4 (*Nix et al., 2008*). Transcripts with a log2 ratio of 1 or higher at a false discovery rate (FDR) $\leq$ 0.05 were considered differentially expressed. Gene enrichment analysis was performed using GOrilla (*Eden et al., 2009*), which determines enriched GO terms at the top of a ranked list of genes. BioProspector (*Liu et al., 2001*) was used for de novo motif finding with the width parameter set to 16. Consistent motifs were identified using 1000 bp upstream of upregulated and downregulated genes.

## Pathogen infection and lifespan analysis

*P. aeruginosa* strain PA14 slow-kill (SK) assays were performed as previously described (*Tan et al., 1999*). PA14 was cultured in Luria broth and seeded on slow-kill plates. Eggs from gravid adults were obtained by alkaline hypochlorite treatment, allowed to hatch overnight in S-basal medium, and arrested in the L1 stage. Synchronized L1 stage worms were grown on NGM plates seeded with OP50 until reaching the L4 stage. Each strain of L4 stage worms were transferred to NGM or SK

plates containing 75 uM 5-fluoro-2-deoxyuridine (FUdR) to prevent growth of progeny and incubated at 22°C. *C. elegans* viability was scored every day using a dissecting microscope to detect dead worms (*Figure 8—source data 1*). For iron content quantification, worms were harvested after 24 h PA14 exposure and processed for ICP-MS analysis.

Lifespan analysis of *nhr-14(tm1473)* and wild-type N2 worms grown under low iron conditions was performed by transferring L4 stage worms to NGM and NGM-25 uM BP plates containing 75 uM FUdR to prevent growth of progeny. Assays were carried out at 22°C and worms were scored every day. For pathogen and lifespan analysis, worms were scored as dead if they did not move, pump, or respond to gentle prodding. Survival analyses were analyzed by Kaplan-Meir method, and statistical significance was assessed using the Mantel-Cox log-rank test (GraphPad Prism).

## Western blot analysis

At indicated time points, worms were washed off the plate with $ddH_2O$ and washed three times to remove bacteria. Worms were resuspended in lysis buffer (20 mM HEPES pH 7.5, 25 mM KCl, 0.5% NP-40) and disrupted by two 5 s pulses using an Ultrasonic Processor (Sonics) at 50% amplitude. Protein concentrations were determined using Coomassie Plus Protein Assay Reagent (Thermo Scientific). Protein samples were incubated for 10 min at 95°C in NuPage LDS Sample Buffer (Invitrogen) containing 10 mM DTT then subjected to electrophoresis using NuPage 4–12% Bis-Tris gels (Invitrogen) and transferred to Amersham Hybond ECL nitrocellulose (GE Healthcare). Blots were incubated with mouse anti-FLAG antibody (Sigma cat# F3165) along with monoclonal β-tubulin antibody (MP Biomedicals LLC cat# 08691261) in 1x Tris-buffered saline (0.01% Tween-20) containing 5% non-fat dry milk. Horseradish peroxidase-conjugated secondary antibodies (Jackson Laboratories) were visualized using Western Lighting Plus-ECL Chemiluminescence Substrate (Perkin Elmer). Three biological replicates were performed and NHR-14::GFP::FLAG was quantified by densitometry analysis using Image J (NIH).

## Iron content quantification

Synchronized L1 stage worms were grown on NGM plates seeded with OP50 until reaching the L4 stage. Worms were washed extensively with M9 buffer and incubated in M9 buffer at room temperature for 2 hr to allow for purging of the gut followed by three rinses with $ddH_2O$. Worms were pelleted and dried, and metal analysis determined by inductively-coupled plasma-optical emission spectroscopy (ICP-OES) (Children's Hospital Oakland Research Institute (CHORI) Elemental Analysis Facility). Empty tubes were run in parallel to serve as controls. Iron content was normalized to sulfur content. For pathogen experiments, worms were harvested in water, washed and homogenized for metal analysis. Protein concentration in lysates was measured using the BCA Protein Assay (Thermo Fisher Scientific) and 100 µg of lysate was digested overnight in 5:1 $HNO_3$:$H_2O_2$, dried, and the pellet resuspended in 2% $HNO_3$ for metal analysis by inductively-coupled plasma mass spectrometry (ICP-MS) (University of Utah, Center for Iron and Hematology Disorders, Iron and Heme Core). At least 3–5 biological experiments were performed for each strain with triplicate technical replicates. Calibration standard solutions for determination of Fe, Zn, Cu and Mn were prepared from Agilent multi-element calibration standard-2A. Iron content was normalized to protein.

## COPAS biosort

Animals expressing P*smf-3::GFP-H2B* containing 1,500 bp of *smf-3* promoter sequences (*Romney et al., 2011*), P*smf-3(mDAE1)::GFP-H2B*, P*smf-3(mDAE2)::GFP-H2B* and P*smf-3(mDAE3)::GFP-H2B* reporters were synchronized by hypochlorite treatment, and arrested L1 stage larvae were grown on 10 cm NGM plates containing *nhr-14* RNAi or control (empty vector) RNAi for 48 hr described above. Worms were washed in M9 buffer and transferred to *nhr-14* RNAi or control RNAi plates supplemented with either 100 µM BP or 6.6 mg/mL FAC and grown overnight. Worms were collected by centrifugation and washed several times in M9 buffer to remove bacteria and debris. Worms were analyzed using COPAS Biosort where L4 stage to young adult worms were gated based on extinction and time of flight parameters (TOF). Extinction and TOF parameters were held constant for subsequent GFP fluorescence acquisition throughout all conditions. GFP fluorescence for 1000 worms was analyzed using FlowJo.

## Microscopy

Transgenic worms carrying NHR-14::GFP::FLAG fosmid were imaged at 40x and 60x using an Olympus FV1000 confocal microscope and camera using Olympus FV10-ASW 3.1 confocal imaging software. PQM-1::GFP::FLAG worms were imaged at either 20x or 40x using a Leica DM6000B microscope. Following acquisition, images were rotated, cropped and sized using Adobe Photoshop.

## Statistical analysis

Statistical analyses were performed using GraphPad software Prism 6 or 7.04 and Excel. Results are expressed as the mean ± SEM. Data were analyzed by an unpaired two-tailed Student's *t*-test or two-way ANOVA with Tukey's multiple comparison test. Survival comparisons were performed using the Mantel-Cox log-rank test. All data were evaluated at the significance level $p \leq 0.05$. Biological replicates reflect different sources of material and/or experiments performed on different days. Statistical details for experiments are indicated in the figure legends.

# Acknowledgements

This work was supported by the NIH awards R01DK068602 (EAL), T32DK007115 (CPA and PMR) and R00HG006922 (JG) and University of Utah SEED grant (EAL). This work was supported by the NIDDK Cooperative Hematology Specialized Core Center award U54DK110858, specifically the Iron and Heme Core for assistance with metal analysis, and by the High-Throughput Genomics and Bioinformatic Analysis Shared Resource at Huntsman Cancer Institute at the University of Utah NIH award P30CA042014. Strains were provided by the *Caenorhabditis* Genetics Center (CGC) and *nhr-14 (tm1473)* was obtained from the National Bioresource Project, Tokyo Womens Medical University. NHR-14::GFP::FLAG fosmid was provided by TransgeneOme. We thank Ms. Diana Lim for figure preparation and the reviewers for comments and suggestions. The authors have declared that no competing interests exist.

# Additional information

## Funding

| Funder | Grant reference number | Author |
|---|---|---|
| NIH Office of the Director | R01DK068602 | Elizabeth A Leibold |
| NIH Office of the Director | T32DK007115 | Cole P Anderson Paul M Rindler |
| NIH Office of the Director | R00HG006922 | Jason Gertz |

The funders had no role in study design, data collection and interpretation, or the decision to submit the work for publication.

## Author contributions

Malini Rajan, Paul M Rindler, Conceptualization, Formal analysis, Validation, Investigation, Visualization, Methodology, Writing—review and editing; Cole P Anderson, Conceptualization, Formal analysis, Validation, Investigation, Visualization, Methodology, Writing—original draft; Steven Joshua Romney, Conceptualization, Formal analysis, Validation, Investigation, Methodology, Writing—review and editing; Maria C Ferreira dos Santos, Formal analysis, Investigation, Writing—review and editing; Jason Gertz, Software, Formal analysis, Methodology; Elizabeth A Leibold, Conceptualization, Formal analysis, Supervision, Funding acquisition, Validation, Investigation, Visualization, Methodology, Writing—original draft, Project administration, Writing—review and editing

## Author ORCIDs

Malini Rajan ⓘ https://orcid.org/0000-0002-7653-4223
Elizabeth A Leibold ⓘ https://orcid.org/0000-0003-1000-9503

Decision letter and Author response
Decision letter https://doi.org/10.7554/eLife.44674.028
Author response https://doi.org/10.7554/eLife.44674.029

## Additional files

### Supplementary files

• Supplementary file 1. Table S1. *C. elegans* strains used in this study. This table lists worm strains generated in this study as well as previously referenced strains.
DOI: https://doi.org/10.7554/eLife.44674.023

• Transparent reporting form
DOI: https://doi.org/10.7554/eLife.44674.024

### Data availability

RNA-seq data have been deposited in GEO under accession code GSE89783. In addition, raw RNA-seq data are reported in the source data files.

The following dataset was generated:

| Author(s) | Year | Dataset title | Dataset URL | Database and Identifier |
|---|---|---|---|---|
| Rajan M, Anderson CP, Rindler PM, Romney SJ, Ferreira dos Santos MC, Gertz JL, Leibold EA | 2019 | NHR-14 loss of function couples intestinal iron uptake with innate immunity in *C. elegans* through PQM-1 signaling | http://www.ncbi.nlm.nih.gov/geo/query/acc.cgi?acc=GSE89783 | NCBI Gene Expression Omnibus, GSE89783 |

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
