## [Decision Letter]

[Editors’ note: a previous version of this study was rejected after peer review, but the authors submitted for reconsideration. The first decision letter after peer review is shown below.]

Thank you for submitting your work entitled "NHR-14 integrates iron uptake and innate immunity in *Caenorhabditis elegans* through PQM-1 signaling" for consideration by *eLife*. Your article has been reviewed by three peer reviewers, and the evaluation has been overseen by a Reviewing Editor and a Senior Editor.

Our decision has been reached after consultation between the reviewers. Based on these discussions and the individual reviews below, we regret to inform you that your work will not be considered further for publication in *eLife*.

All the reviewers found the manuscript to be very interesting, but the lack of rigor or in-depth analysis in many areas makes it unsuitable for publication in its current form. The manuscript could be improved by going more in depth on the metal biology or by strengthening the innate immunity aspects. The diversity of the reviewer's suggestions indicate that the revision will likely take more than two months, the time limit in e*Life* for revision. This is why the paper was rejected. Nevertheless, the authors could consider re-submitting their article to *eLife* as a new submission once the reviewer's comments have been addressed in full.

Reviewer #1:

The authors had previously discovered a role for the *C. elegans* HIF1a homolog in regulating intestinal iron homeostasis during iron deficiency by activating and inhibiting genes involved in iron uptake and storage. Taking advantage of a developmental delay phenotype of *hif-1(ia4)* mutants on low iron media, the authors performed a suppressor screen and uncovered two mutations in a nuclear hormone receptor gene, *nhr-14*, that rescued the delay. NHR-14 transcript and protein levels were unaffected by iron levels. RNA-seq anaylsis comparing gene expression profiles between wildtype animals and *nhr-14* mutants revealed that *nhr-14* regulates the iron transport gene *smf-3*, and innate immunity genes. Analysis of the *smf-3* promoter and mining modENCODE ChIP-seq data suggested that PQM-1 acts downstream of NHR-14 to regulate *smf-3*. The authors explored the role of NHR-14 in innate immunity, demonstrating that in addition to regulating immune genes, *nhr-14* mutants were resistant to P. aeruginosa and *S. enterica* infection, and that bacterial infection downregulated NHR-14 protein levels. Finally, the authors showed that knockdown of the *nhr-14* ortholog, HNF4a, in human cells resulted in a similar upregulation of the *smf-3* homolog (DMT1). This is an interesting study looking at how nuclear hormone receptors coordinate iron availability and innate immunity, but the manuscript suffers from a lack of details in the genetic analyses, and a lack of exploration of the roles of NHR-14 in innate immunity required to show that the gene expression signature is not simply a consequence of disrupting a core cellular process.

Substantive concerns

1) Elements of the manuscript were difficult to evaluate, as many details about the screen and the genetics were lacking. For the screen, how many genomes were screened? How many mutants were recovered? Was the screen saturating? As this was the first report of a screen, these details should be provided. Additionally, were the suppressors, and the *nhr-14(tm1473) pqm-1(ok485)* mutants backcrossed? If so, how many times?

2) There is a complete lack of characterization of the *nhr-14* mutant strains that hampers evaluation of many elements of the manuscript. Do these strains have normal broodsizes? Do they develop at a normal rate? Do they have a normal lifespan? These questions are particularly important for interpreting the pathogen data. What is the *nhr-14(tm1473)* allele? Is it a null? Are the three *nhr-14* mutants phenotypically equal? If so, that would be particularly interesting as the insertion in the DNA binding domain would be predicted to disrupt DNA binding, but not the activation domains or ligand binding domain, and would support that direct gene regulation by NHR-14 contributes to the suppression of the *hif-1a* arrest on low iron media.

3) The innate immunity sections are problematic. Melo and Ruvkun (2012) demonstrated that RNAi disruption of core cellular processes induces expression of detoxification and innate immune effectors, even in the absence of toxins or pathogens. Similar to point 1, much more information about the *nhr-14* strains needs to be provided to convince the reader that this is specifically an immune phenotype and not due to affecting a core process. The authors claim that NHR-14 is the first *C. elegans* NHR to be involved in innate immunity (and iron metabolism), but Ward et al., 2014, demonstrated that inactivation of nhr-25 causes sensitivity to a range of stresses and pathogens. Several experiments would help strengthen the immunity sections: i) lifespan on OP50 as a baseline for comparing the pathogen sensitivity data; ii) NHR-14 overexpression, which would be predicted to cause increased sensitivity to pathogens. Kirienko et al., 2013, reported a liquid killing assay for PA14 that involves different pathogenesis mechanisms than plate-based assays and is particularly dependent on iron. This assay would be very useful to evaluate the importance of NHR-14-mediated iron regulation in the context of immunity.

4) A major weakness of the paper is lack of any mechanism of how NHR-14 is regulating PQM-1 and in turn SMF-3, or how pathogenic stress regulates NHR-14 levels. An INS-7 overexpression experiment would be very straightforward.

Reviewer #2:

Elucidating mechanisms that organisms use to mediate iron homeostasis through the regulation of transcription factors and transport proteins is an important objective in the field of metal biology. This manuscript details the interaction of two transcription factors that link iron homeostasis with the response of *C. elegans* to infection by pathogenic bacteria. The authors performed an innovative forward genetic screen leading to the identification of two mutations in the *nhr-14* gene. They used rigorous methods to demonstrate that mutations in *nhr-14* cause the phenotype. They characterized the phenotype of *nhr-14* mutant animals, including the rescue of the hypersensitivity to low iron phenotype of the *hif-1* mutant, an increase in metal content for iron and manganese, an increase in the expression of many genes including *smf-3* and multiple genes linked to innate immunity, and an alteration in the localization of the transcription factor PQM-1, and resistance to bacterial infections. They also demonstrate that the loss of the human homolog of NHR-14 causes an increase in the expression of another iron transport protein in vertebrate cells, suggesting a conservation of function of the identified nuclear receptor. The impact and quality of the work found within this manuscript would make it a strong candidate for *eLife* if the authors address the following concerns:

Major Issues:

1) The authors identified mutations in *nhr-14* as suppressors of the low iron hypersensitivity of *hif-1*, which is an innovative discovery technique. They used powerful positional cloning approaches to clearly demonstrate that the mutations in *nhr-14* are the cause of the enhanced growth in low iron. The major question that arises is how does mutation of *nhr-14* rescue the *hif-1* growth defect, and a key approach is to analyze the phenotype of *nhr-14* mutations in an otherwise wild type background. This manuscript would benefit from experiments that compare the growth of *nhr-14* mutants to wild type animals in two or three concentrations of BP and two or three concentrations of FAC. If the *nhr-14* mutations do indeed act by enhancing iron uptake, then it is predicted that they will enhance growth in BP compared to wild-type animals, and they may be hypersensitive to FAC compared to wild type animals. If these effects are not observed, it would call into question the model that the *nhr-14* mutation acts by enhancing iron uptake.

2) The authors argue that the cellular site of action of *nhr-14* is intestinal cells, as shown in Figure 9. This conclusion appears to be based on the localization of a NHR-14:GFP fusion protein, as shown in Figure 2D. However, the fusion protein is also expressed in other cell types – at a minimum, the red arrows highlight expression in head neurons. It is not possible to judge from the single image how many other cell types may also express NHR-14. Thus, if the conclusion about cellular site of action is based solely on expression data, then an alternative possibility is that the key site of action is the head neurons, or other cell types that may not be visible in this single image. If the authors want to draw a conclusion about the cellular site of action, then they should do experiments to express NHR-14 in a specific cell type using a heterologous promoter, such as an intestinal specific promoter. I do not think determining the cellular site of action is essential for the main conclusions of the paper, so another approach is for the authors to deemphasize the conclusion that the site of action is intestinal cells.

3) The authors investigated the exciting hypothesis that *nhr-14* is regulated by dietary iron. However, the results shown in Figure 2 are negative and support the conclusion that both the transcriptional and translational levels of NHR-14 are unaffected by dietary iron. However, a possible unexplored explanation for the function of NHR-14 in iron homeostasis is that the localization of NHR-14 is altered in response to changes in supplemental iron, since some nuclear receptors are regulated by nuclear localization. The functional characterization of NHR-14 would be more complete with an examination of the localization change when animals expressing the tagged version of NHR-14 are exposed to excess and limited iron conditions.

4) The conclusion that *nhr-14* acts through regulation of *smf-3* to increase the growth of *hif-1* mutants should be considered more rigorously. The authors clearly demonstrate that *nhr-14* mutants have higher levels of *smf-3* transcript, and SMF-3 is a known iron import protein, leading to the hypothesis that this is the critical target gene for the *hif-1* rescue. I acknowledge this is a reasonable and even plausible hypothesis. However, the transcriptional analysis of *nhr-14* mutant animals demonstrates that hundreds of transcripts are altered, so in principle any one of these changes, or many of these changes in combination might mediate the *hif-1* rescue. The critical issue is what additional evidence can test the *smf-3* hypothesis. The authors create a *smf-3; nhr-14; hif-1* triple mutant and the rescue is lost, in that these triple mutants do not grow in low iron. My concern is that the *smf-3* single mutants may not be able to grow in low iron, in which case this experiment is not really informative. For example, if the authors create a triple mutant with *nhr-14, hif-1* and any mutation that causes L1 arrest, they will get this result. To interpret this experiment, the authors should analyze the *smf-3* single mutant in BP. If the *smf-3* single mutant displays arrested growth, then I think there is no clear interpretation of the double mutant of *smf-3* with *hif-1* and the triple mutant of *smf-1, hif-1* and *nhr-14*. These data would therefore not support the conclusion that NHR-14 functions in iron homeostasis through *smf-3*.

5) In Figure 3B, there is a significant increase in the amount of manganese in animals with a mutation in *nhr-14*. This raises the possibility that the change in manganese levels causes any or all of the *nhr-14* mutant phenotypes. It would be helpful if the authors addressed this possibility, certainly in the text and possibly experimentally. Have the authors examined the phenotypes of *nhr-14* mutants with respect to excess and limited manganese?

6) The model that NHR-14 works in combination with PQM-1 to reduce the iron in the intestinal lumen through SMF-3 is speculative. For example, there are no measurements of the level of iron in the intestinal lumen. Alternative models are that it is changes in iron in the body of the worm that influence pathogen resistance, or that there is a non-iron based mechanism of pathogen resistance. One possible test of the model could be examining the effect of adding supplemental iron to the pathogenicity of the tested bacteria. This would help to test the prediction that decreasing the iron in the lumen is protective to these animals. In a similar vein, the model suggests that *nhr-14* mutant animals are resistant because there is an increase in the expression of *smf-3* in these animals. What are the levels of *smf-3* expression in mutant animals exposed to pathogenic bacteria? This could be included as part of Figure 7D and 7F. These data would also provide support for the current model.

7) Looking at the pathogen resistance data, it was surprising that the authors decided to pursue further phenotype analysis of the *nhr-14* mutant using PA14 rather than *S. enterica* as the pathogenic bacteria tested. The lifespan phenotype associated with *S. enterica* was much stronger. The data would be more compelling if Figure 7I was performed with *S. enterica*. Also in Figure 7I, the intermediate phenotype of the double mutant is hard to interpret and does not support the idea that NHR-14 acts upstream of *smf-3*. If the loss of NHR-14 makes animals resistant through SMF-3, then the double mutant of *nhr-14* and *smf-3* would be expected to be as hypersensitive as the single mutant of *smf-3*.

Reviewer #3:

The paper is well written, the methods are very diverse and appropriately used, and the approach is sound and thorough. The topic is trendy, the interpretations and conclusions are fair and the scientific contribution is significant.

The authors describe a new role for the transcription factor NHR-14, attracting attention to the somewhat understudied extended family of 269 *C. elegans* HNF4-derived transcription factors. They find a key role for NHR-14 in gut infection handling through iron status modulation (intake by the DMT1/NRAMP2 orthologue SMF-3 and sequestration by FTN-1) and transcriptional regulation of a set of innate immunity-associated genes. Notably, they also identify PQM-1 has a downstream effector that shuttles between the cytosol and the nucleus of enterocytes to directly regulates *smf-3* expression.

This study hence connects PQM-1, which is gaining recognition as a major transcription factor in the modulation of *C. elegans* fat handling, iron metabolism and ageing, to innate immunity. Although expected and supported by two other recent studies, it is an important step toward an understanding of *C. elegans* innate immunity regulation. Moreover, the upstream role of NHR-14 could not have been inferred from the current literature and it contributes significantly to piecing together the complex network that regulates trade-offs between energy handling and immune responses.

There is a great potential for further studies trying to delineate the exact role of NHR-14 in resistance to infection (it seems specific to certain infections), its upstream regulators (there are hints from mammalian studies, notably specific fatty acids) and how it modulates PQM-1 shuttling, and indirectly DAF-16 shuttling. Connections between immunity, fat metabolism, lipid signalling and aging are being noticed more and more and this study contributes to solving the puzzle.

On a more technical note, I liked the fact that the authors made the effort to accumulate corroborative evidence without relying abusively on a single technical approach. The genetic work is also appreciated (double and triple mutants will make excellent tools for the community). I also liked a couple of points of detail that helped convince me of the interpretations. For instance, the qRT-PCR results give fold increases that match very well the fold increases seen by RNAseq. Also, in Figure 7I, *nhr-14* seems to be able to increase *smf-3* resistance to P. aeruginosa infection, which is consistent with the idea that NHR-14 increases infection resistance through other genes than *smf-3* and iron sequestration alone.

General concerns.

A couple of side issues may have been neglected to ensure that the picture presented is not blown out of proportions (see specific concerns).

The raw and analysed data from the RNAseq and lists of overlapping genes between different datasets should be available. If not, the claims of the authors cannot be verified. It is also common practice to make these datasets available upon publication. It would help the peer-review process if at least the analysed data used in the manuscript were made available to the reviewers.

Although the methods are generally sound, some should be expanded upon to allow for replication of the results. Numbers of animals and/or replicates are low for some experiments and would require additional repeats to meet desirable standards. When it comes to biological replicates, their definition is unclear. One would expect that each biological replicate would be a different population on a different day as interday variability is far greater for *C. elegans* populations compared to intraday variability, but the low numbers might indicate that biological replicates came from the same day. Clarification is required on this issue.

Specific concerns.

First, the title is slightly misleading. Although the authors provided evidence that a NHR-14 – PQM-1 axis does integrate elements of iron metabolism through SMF-3, the case is less strong when it comes to innate immunity. Indeed, no clear immunity pathway has been defined as it is mostly based on observed enrichment in innate immunity-associated genes in RNAseq analysis without hierarchy. Moreover, direct regulation of innate immunity genes by PQM-1 has not been verified (using biochemical assays, qRT-PCR under infection conditions, or transcriptional reporters in pqm-1 mutant, and/or WT animals under infection conditions). "Integrate" is a too strong word as it suggests pivotal role that is not demonstrated.

Second, SMF-1 and SMF-2 are two other DMT1 homologues in *C. elegans* and SMF-1 is expressed in the gut just as SMF-3 is. Yet there is no comment on SMF-1 potential role here and its possible regulation by *nhr-14*, which should have been checked and reported.

The specificity of NHR-14 action on PQM-1 is not clearly established especially in the absence of pathway between the two. Several other high-ranking intestinal transcription factors (ELT-2, DAF-16, SKN-1, ATF-7, NHR-49) have been involved in gut immunity in worms and could have been looked at more carefully, especially when transgenic reporters are available to look at nuclear shuttling (SKN-1::GFP and DAF-16::GFP for instance).

The role of pqm-1 in infection resistance is not clearly established, while *nhr-14* and *smf-3* are.

HIF-1 is mentioned initially because *nhr-14* was found in a suppressor screen for the *hif-1* growth defect under iron deprivation, but it is not discussed later, nor is it connected to the findings.

Suggested additional experiments.

– Infections assays with P. aeruginosa and *S. enterica* on pqm-1 mutants are necessary to demonstrate a functional role of PQM-1 in innate immunity (high priority).

– To corroborate this the PQM-1::GFP strain should be exposed to P. aeruginosa and nuclear shuttling should be observed (high priority).

– Make sure there are at least 3 independent experiments (different days) with a minimum of 30 worms per strain for the infection assays. A good idea would be to simply do the additional experiments required with nhr-14, WT and pqm-1 mutants in parallel (high priority).

– *C. elegans* gut bacterial infections are performed at adult stage because larvae usually do not take up bacteria. It is essential that the PQM-1::GFP nuclear localisation shift seen in L3 larvae on *nhr-14* RNAi is replicated in adult worms to convince that PQM-1 can act upon gut bacterial infection to regulate innate immunity genes (high priority).

– qRT-PCR of smf-1, -2 and -3 in *nhr-14* and in pqm-1 at least (double mutants *nhr-14*;pqm-1 welcome) to demonstrate the specificity of SMF-3 regulation (if time and money are an issue, it is not the highest priority experiment)

– qRT-PCR of pqm-1 in *nhr-14* mutant vs WT to confirm the RNAseq data (low priority, up to the good will of the authors)

– Look at DAF-16::GFP vs PQM-1 shuttling in control, *nhr-14*, and pqm-1 RNAi to confirm the antagonistic shuttling between DAF-16 and PQM-1 and the specificity of the effect of *nhr-14* on PQM-1 vs DAF-16 or ELT-2 (medium priority, depending on which other experiments are done)

– P. luminescens, E. faecalis, S. marscens, S. aureus infections would have made sense in Figure 7 given that the authors compare their RNAseq data to datasets on worms infected by these pathogens. This seems particularly valid as Table 2 seems to point toward a more specific role of *nhr-14* downregulation in resistance against certain pathogens and not others (low priority, up to the good will of the authors).

– Inclusion of the elt-2 microarray dataset published by Block DH et al., 2015 would make sense in Figure 6 and Table 2, considering that the present article did briefly investigate elt-2. (medium priority).

[Editors’ note: what now follows is the decision letter after the authors submitted for further consideration.]

Thank you for submitting your article "NHR-14 loss of function couples intestinal iron uptake with innate immunity in *C. elegans* through PQM-1 signaling" for consideration by *eLife*. Your article has been reviewed by three peer reviewers, and the evaluation has been overseen by a Reviewing Editor and Tadatsugu Taniguchi as the Senior Editor. The reviewers have opted to remain anonymous.

The reviewers have discussed the reviews with one another and the Reviewing Editor has drafted this decision to help you prepare a revised submission.

Summary:

Elucidating mechanisms that organisms use to mediate iron homeostasis through the regulation of transcription factors and transport proteins is an important objective in the field of metal biology. This manuscript details the interaction of two transcription factors that link iron homeostasis with the response of *C. elegans* to infection by pathogenic bacteria. The authors performed an innovative forward genetic screen leading to the identification of two mutations in the *nhr-14* gene. They used rigorous methods to demonstrate that mutations in *nhr-14* cause the phenotype. They showed NHR-14 is expressed in intestinal cells and other cells in the head and does not appear to be regulated by iron. They characterized the phenotype of *nhr-14* mutant animals, including the rescue of the hypersensitivity to low iron phenotype of the *hif-1* mutant, an increase in metal content for iron, an increase in the expression of many genes including *smf-3* and multiple genes linked to innate immunity, nuclear localization of PQM-1, and resistance to bacterial infections. PQM-1 appears to interact directly with the *smf-3* promoter. The findings lead to the proposal that pathogenic infections down regulate nhr-14, leading to increased expression of SMF-3, which imports iron into intestinal cells, thereby depriving the pathogen of iron in the lumen of the intestine and limiting its capacity for proliferation. While largely conclusions are supported by the results, some additional experiments are needed to reinforce the major findings.

Essential revisions:

1) Transcriptome analysis. A main concern is with the analysis presented in Figure 5. Apart from the fact that DAVID has not been updated (https://david.ncifcrf.gov/helps/update.html), it contains far less information than *C. elegans*-specific tools such as WormExp. Running their list through WormExp reveals several very significant overlaps with other datasets, including nhr-8 mutants that merit at least discussion if not investigation.

Further, the authors write, "Among the 261 downregulated genes, there was enrichment in genes involving cellular organization and body morphogenesis, including sqt-1, noah-1 and sym-1 (Figure 5D and Figure 3—source data 1)".

But what is really striking is the number of genes that normally undergo a sharp drop of expression at the L4 to adult transition, many of them expressed in the epidermis. As can be visualized at https://elegans.mdc-berlin.de/cel_ex.html, this applies, for example, to sqt-1, col-17, rol-8 (the top 3 in Figure 5D), as well as to the heterochronic gene lin-42. There is no comment on this very remarkable pattern, nor the simple experiments that might address the question of how it might be related to the continued nuclear localization of PQM-1 into adulthood.

On the same line, the authors write, "Interestingly, we observed less overlap between *nhr-14* upregulated genes with *Serratia marscens* [marcescens] (Wong et al., 2007). The basis for this difference is unknown, but is likely a result of a highly specialized response to *S. marscens* [marcescens]". If they had used WormExp, they would have seen a very significant overlap with Serratia-induced genes in independent study (Engelmann et al., 2011). And reading that study, there is even the following, "In the current analysis, the results of our previous study of the response to S. marcescens using oligo-arrays [22] stood out. As this was not the case for the results for the response to 2 other bacterial pathogens, and given the underrepresentation in the S. marcescens data set of 'common response genes' [22], this presumably reflects an experimental difference in the strength of the infection for the samples prepared for analysis using oligo-arrays". In other words, the authors' conclusion is unlikely to be correct. And that makes more sense since the screens that used *C. elegans* to identify S. marcescens virulence factors highlighted a role for iron (see for example the Abstract of PMID: 12660152, and 25070509).

2) The authors clearly demonstrate that *nhr-14* mutants have higher levels of *smf-3* transcript, and SMF-3 is a known iron import protein, leading to the hypothesis that this is the critical target gene for the *hif-1* rescue. I acknowledge this is a reasonable and even plausible hypothesis. However, the transcriptional analysis of *nhr-14* mutant animals demonstrates that hundreds of transcripts are altered, so in principle any one of these changes, or many of these changes in combination might mediate the *hif-1* rescue. The critical issue is what additional evidence can test the *smf-3* hypothesis. The authors create a *smf-3; nhr-14; hif-1* triple mutant and the rescue is lost, in that these triple mutants do not grow in low iron. My concern is that the *smf-3* single mutants also do not grow in low iron. So, yes *nhr-14* requires *smf-3* to rescue *hif-1*, but this is somewhat different from proving that *nhr-14* rescues by increasing *smf-3* activity. The authors should acknowledge that because the *smf-3* single mutant cannot grow in low iron, there are multiple interpretations for the triple mutant.

3) Additional controls:

3A) Figure 3. Results shown in panels B, C, and D lack essential genotype that has to be tested, namely nhr-14,*smf-3* double mutant. This genotype is essential to prove that the phenotypes of *nhr-14* mutants are mostly mediated by the elevated level of *smf-3*.

3B) Figure 5E and D. It is nice to see enrichment in PQM-1 binding motif in the promoters of the upregulated genes, but experimentally do these genes really require PQM-1 for their expression? What will be their level of induction in nhr-14, pqm-1 double mutant? I think this experiment is important to prove that pqm-1 regulates these genes downstream of nhr-14.

3C) Figure 6A. This experiment lacks nhr-14, pqm-1 double mutant. Again, this is essential to prove epistatic relations between the two genes.

---

## [Author Response]

[Editors’ note: the author responses to the first round of peer review follow.]

All the reviewers found the manuscript to be very interesting, but the lack of rigor or in-depth analysis in many areas makes it unsuitable for publication in its current form. The manuscript could be improved by going more in depth on the metal biology or by strengthening the innate immunity aspects. The diversity of the reviewer's suggestions indicate that the revision will likely take more than two months, the time limit in eLife for revision. This is why the paper was rejected. Nevertheless, the authors could consider re-submitting their article to eLife as a new submission once the reviewer's comments have been addressed in full.

Based on the reviewers’ suggestions and comments, we made substantial changes to the manuscript, including addition of new iron pathogen experiments, reanalysis of RNA-seq data and addition of genetic details, which has resulted in a stronger manuscript.

In this manuscript, we report on the identification of a relatively uncharacterized nuclear receptor-1 (NHR-14) that couples innate immunity with iron sequestration. NHR-14 shares homology with vertebrate HNF4. We identified *nhr-14* in a genetic suppressor screen conducted to identify genes that rescues the developmental delay of *hif-1* mutants grown under iron limitation. We found that *nhr-14* loss of function rescues the *hif-1* developmental phenotype through increased expression of the intestinal iron importer SMF-3. Transcriptome studies using *nhr-14* mutants revealed enrichment of innate immune response genes, many of which are Class 2 genes reported by Murphy et al. (Nature 2003) to be repressed by DAF-16/FOXO during reduced insulin signaling. More recently, Tepper et al. (Cell 2011) discovered that Class 2 genes were activated by the PQM-1 transcription factor during reduced insulin signaling. Using genetic and biochemical approaches, we showed that *smf-3* is a PQM-1 target gene. Given the innate immune response and iron signatures in *nhr-14* mutants, we showed that *nhr-14* loss of function is required for pathogen resistance and that iron sequestration is a component of the pathogen innate immune response. Several nuclear receptors are known regulators of innate immunity in *C. elegans*; however, NHR-14 is unique among these receptors in coupling innate immunity with iron sequestration. Our data provide new knowledge into how *C. elegans* use nuclear receptors to regulate innate immunity and iron availability, and show iron sequestration as an important component of the innate immune response.

We believe that our work is of current interest as several of the genes in *nhr-14* mutants have been identified in other stress response pathways in *C. elegans.* For example, O’Brien et al., *2018,* identified PQM-1 as a regulator of transcellular chaperone signaling in neurons and in intestine. Several of the key components of neuronal and intestinal signaling pathways and are upregulated in *nhr-14* mutants. Of interest, *nhr-14* expression is also enriched in neuronal and intestinal cells, providing further evidence suggesting a role for NHR-14 in these pathways. In another paper, Jiang, et al. (eLife2018) identified a genetic pathway that regulates hypothermia stress, and several of the key genes in this pathway, including a transcription factor and proteases, are upregulated in *nhr-14* mutants. It is thus likely that NHR-14 has a function in these pathways.

As NHR-14-PQM-1 represents a new innate immunity pathway, future studies are needed to identify other key components in this pathway and endogenous or exogenous ligands that regulate NHR-14 transcriptional function, and to determine the role of NHR-14 in the stress response pathways described above.

[Editors' note: the author responses to the re-review follow.]

Essential revisions:1) Transcriptome analysis. A main concern is with the analysis presented in Figure 5. Apart from the fact that DAVID has not been updated (https://david.ncifcrf.gov/helps/update.html), it contains far less information than C. elegans-specific tools such as WormExp. Running their list through WormExp reveals several very significant overlaps with other datasets, including nhr-8 mutants that merit at least discussion if not investigation.

We thank the reviewer for this suggestion.

We compared upregulated and downregulated *nhr-14* genes with WormExpv1.0 “mutant” category. We found that 148 upregulated nhr-14 genes out of 568 genes overlapped with genes upregulated in *nhr-8* mutants. Also, of interest, 156 upregulated *nhr-14* genes overlapped with hyl-2 (encoding ceramide synthase) mutants. Together, these observations suggest a role for *nhr-14* in fat metabolism. Iron and lipid metabolism are related and many enzymes involved in lipid metabolism require iron for activity. We added the above gene comparisons to a supplemental table (Figure 6—source data 4).

Using WormExp “tissue” category and Cao et al., 2017 single-cell PCR dataset, we found that downregulated *nhr-14* genes are enriched in hypodermis and neurons and *nhr-14* upregulated genes are enriched in intestine and neurons. These data are consistent with enriched expression of the NHR-14::GFP::FLAG transgene in intestine and head cells (Figure 3B-D). WormExp “tissue” dataset overlap with upregulated and downregulated *nhr-14* genes in reported in Figure 5—source data 1.

Further, the authors write, "Among the 261 downregulated genes, there was enrichment in genes involving cellular organization and body morphogenesis, including sqt-1, noah-1 and sym-1 (Figure 5D and Figure 3—source data 1)".But what is really striking is the number of genes that normally undergo a sharp drop of expression at the L4 to adult transition, many of them expressed in the epidermis. As can be visualized at https://elegans.mdc-berlin.de/cel_ex.html, this applies, for example, to sqt-1, col-17, rol-8 (the top 3 in Figure 5D), as well as to the heterochronic gene lin-42. There is no comment on this very remarkable pattern, nor the simple experiments that might address the question of how it might be related to the continued nuclear localization of PQM-1 into adulthood.

We examined the developmental expression of the top 50 downregulated *nhr-14* genes using MDC-BIMSB. In agreement with this reviewer observations, we found that expression of 42 out of the top 50 downregulated *nhr-14* genes were sharply downregulated at the late L4 stage, while only 3 were upregulated. The 3 upregulated genes were enriched in intestine, while downregulated *nhr-14* genes were enriched in seam and non-seam hypodermal cells (determined from Cao, et al., 2017 single-cell RNA-seq).

We also examined the developmental expression of the top 50 upregulated *nhr-14* genes and found most genes are also downregulated at late L4 (20 genes were unchanged or not in the dataset and 6 were upregulated), but not as sharply as the downregulated *nhr-14* genes. Of interest, the 6 genes upregulated at late L4 were DAF-16/FoxO Class II genes (*dod-19,* dod-*21, dod-23, dod-24, oac-14*, C32H11.9). qPCR analysis revealed that expression of *dod-19, dod-24, oac-14* and *clec-41 (dod -21* and C32H11.9 were not assessed) were reduced in *nhr-14; pqm-1 (RNAi*) mutants, indicating that PQM-1 regulate these genes downstream of *nhr-14.* These new data are reported in Figure 7. See comment 3B below.

We appreciate the reviewer’s effort in uncovering these interesting observations regarding the developmental expression of *nhr-14* genes. We are pursuing how loss of *nhr-14* causes the nuclear accumulation of PQM-1 in adults, and how it affects gene expression and aging. However, we prefer not to discuss the developmental expression of upregulated and downregulated *nhr-14* genes in this manuscript as it would require inclusion of more datasets as well as experiments. These experiments would require a considerable amount of work to do rigorously. We feel that these experiments are beyond the scope of our manuscript, which is to understand the link between *nhr-14* regulation of innate immunity and iron metabolism.

On the same line, the authors write, "Interestingly, we observed less overlap between nhr-14 upregulated genes with Serratia marscens [marcescens] (Wong et al., 2007). The basis for this difference is unknown, but is likely a result of a highly specialized response to S. marscens [marcescens]". If they had used WormExp, they would have seen a very significant overlap with Serratia-induced genes in independent study (Engelmann et al., 2011). And reading that study, there is even the following, "In the current analysis, the results of our previous study of the response to S. marcescens using oligo-arrays [22] stood out. As this was not the case for the results for the response to 2 other bacterial pathogens, and given the underrepresentation in the S. marcescens data set of 'common response genes' [22], this presumably reflects an experimental difference in the strength of the infection for the samples prepared for analysis using oligo-arrays". In other words, the authors' conclusion is unlikely to be correct. And that makes more sense since the screens that used C. elegans to identify S. marcescens virulence factors highlighted a role for iron (see for example the Abstract of PMID: 12660152, and 25070509).

We agree and revised this section. We compared the Engelmann et al., 2011 Serratia dataset with upregulated *nhr-14* genes and found that out of 2124 genes in the Engelmann dataset, 237 overlapped with upregulated *nhr-14* genes. These data are shown in Figure 6A and listed in Figure 6—source data 1. The Wong et al., 2007 Serratia dataset was deleted.

2) The authors clearly demonstrate that nhr-14 mutants have higher levels of smf-3 transcript, and SMF-3 is a known iron import protein, leading to the hypothesis that this is the critical target gene for the hif-1 rescue. I acknowledge this is a reasonable and even plausible hypothesis. However, the transcriptional analysis of nhr-14 mutant animals demonstrates that hundreds of transcripts are altered, so in principle any one of these changes, or many of these changes in combination might mediate the hif-1 rescue. The critical issue is what additional evidence can test the smf-3 hypothesis. The authors create a smf-3; nhr-14; hif-1 triple mutant and the rescue is lost, in that these triple mutants do not grow in low iron. My concern is that the smf-3 single mutants also do not grow in low iron. So, yes nhr-14 requires smf-3 to rescue hif-1, but this is somewhat different from proving that nhr-14 rescues by increasing smf-3 activity. The authors should acknowledge that because the smf-3 single mutant cannot grow in low iron, there are multiple interpretations for the triple mutant.

We agree and added the sentence:”Given that the *smf-3(ok1035)* single mutant alone displays a developmental delay under iron limitation, this suggests that other genes in *nhr-14(tm1473)* mutants might also contribute to the rescue of *hif-1(ia4)* mutants”.

3) Additional controls:3A) Figure 3. Results shown in panels B, C, and D lack essential genotype that has to be tested, namely nhr-14,smf-3 double mutant. This genotype is essential to prove that the phenotypes of nhr-14 mutants are mostly mediated by the elevated level of smf-3.

We agree and included iron content of the *smf-3(ok1035); nhr-14(tm1473)* double mutants (Figure 3B) as well as an image and growth analysis of this mutant on NGM and NGM-BP (low iron) (Figure 3C). We did not include the *smf-3; nhr-14* double mutant to the life span analysis in panel D because of its poor growth under iron limitation.

3B) Figure 5E and D. It is nice to see enrichment in PQM-1 binding motif in the promoters of the upregulated genes, but experimentally do these genes really require PQM-1 for their expression? What will be their level of induction in nhr-14, pqm-1 double mutant? I think this experiment is important to prove that pqm-1 regulates these genes downstream of nhr-14.

We agree and used qPCR to measure expression of several upregulated *nhr-14* genes in *nhr-14(tm1473); pqm-1 (RNAi)* and *nhr-14(tm1473*); Control RNAi worms. qPCR analysis showed reduced expression of Class II genes *dod-19, dod-24, clec-41, gst-38* and *oac-14* in *nhr-14; pqm-1 (RNAi*) mutants vs *nhr-14*; Con (*RNAi*) worms. The expression of Class II genes *ins-7* and *lys-2* and Class I genes *lys-7* and *ftn-1* were not changed in *nhr-14; pqm-1 (RNAi*) mutants. These data are shown in Figure 7. See response to comment #1.

3C) Figure 6A. This experiment lacks nhr-14, pqm-1 double mutant. Again, this is essential to prove epistatic relations between the two genes.

We agree and measured the survival *nhr-14(tm1473); pqm-1 (RNAi)* worms after infection with PA14. We found that the *nhr-14(tm1473); pqm-1 (RNAi*) mutants are more sensitive to PA14 that the *nhr-14* single mutant, suggesting that PQM-1 is required for *nhr-14(tm1473*) resistance. These data are shown in Figure 8A (graph) and Figure 8—source data 1 (survival analysis).